

# A method for estimating localized $CO_2$ emissions from co-located satellite $XCO_2$ and $NO_2$ images

Blanca Fuentes Andrade[1], Michael Buchwitz[1], Maximilian Reuter[1], Heinrich Bovensmann[1],
Andreas Richter[1], Hartmut Boesch[1], and John P. Burrows[1]

[1]Institute of Environmental Physics (IUP), University of Bremen, Bremen, Germany

**Correspondence:** Blanca Fuentes Andrade (bfuentes@iup.physik.uni-bremen.de)

**Abstract.**

Carbon dioxide ($CO_2$) is the most important anthropogenic greenhouse gas. Its atmospheric concentration has increased by almost 50% since the beginning of the industrial era, causing climate change. Fossil fuel combustion is responsible for most of the atmospheric $CO_2$ increase, which originates to a large extent from localized sources such as power stations. Independent

estimates of the emissions from these sources are key to tracking the effectiveness of implemented climate policies to mitigate climate change. We developed a procedure to quantify $CO_2$ emissions from localized sources based on a cross-sectional mass-balance approach and applied it to infer $CO_2$ emissions from the Bełchatów Power Station, in Poland, using atmospheric observations from the Orbiting Carbon Observatory 3 (OCO-3) in its Snapshot Area Map (SAM) mode. As a result of the challenge of identifying $CO_2$ emission plumes from satellite data with adequate accuracy, we located and constrained the

shape of emission plumes using TROPOspheric Monitoring Instrument (TROPOMI) $NO_2$ column densities. We analysed all available OCO-3 overpasses over the Bełchatów Power Station from July 2019 to November 2022 and found a total of 9 that were suitable for the estimation of $CO_2$ emissions using our method. The mean uncertainty of the obtained estimates was 5.8 $Mt\,CO_2y^{-1}$ (22.0%), mainly driven by the dispersion of the cross-sectional fluxes downwind of the source, e.g. due to turbulence. This dispersion uncertainty was characterized using a semivariogram, possible thanks to the OCO-3 imaging

capability over a target region in SAM mode, which provides observations containing plume information up to several tens of kilometres downwind of the source. A bottom-up emission estimate was computed based on the hourly power plant generated power and emission factors to validate the satellite-based estimates. We found that the two independent estimates agree within their $1\sigma$ uncertainty in 8 out of 9 analysed overpasses and have a high Pearson's correlation coefficient of 0.92. Our results confirm the potential for monitoring large localized $CO_2$ emission sources from space-based observations and the usefulness

of $NO_2$ estimates for plume detection. They illustrate as well the potential to improve $CO_2$ monitoring capabilities with the planned Copernicus Anthropogenic $CO_2$ Monitoring (CO2M) satellite constellation, which will provide simultaneously retrieved $XCO_2$ and $NO_2$ maps.





## 1 Introduction

$CO_2$ is the most important anthropogenic greenhouse gas and its cumulative atmospheric concentration increase plays a major
role in global warming and climate change (Chen et al., 2021). In 2015, the Paris Agreement was adopted to limit global warm-
ing to well below 2°C and pursue "efforts to limit the temperature increase to 1.5°C above pre-industrial levels" (UNFCCC,
2015). To meet these objectives, net greenhouse gas emissions need to be rapidly reduced (IPCC, 2023; Rockström et al.,
2017). Under this agreement and as part of the mitigation strategy, the parties report their national greenhouse gas inventories,
usually computed using bottom-up methods based on statistical activity data and emission factors (IPCC, 2006). Top-down ap-
proaches, based on atmospheric observations, can complement these inventories and verify their accuracy (Bergamaschi et al.,
2018).

Most of the $CO_2$ emissions result from the combustion of fossil fuels. About one-third of the total fossil fuel emissions
happen at localized sources, such as power plants (Oda and Maksyutov, 2011; IEA, 2019; Crippa et al., 2022). Therefore,
monitoring the $CO_2$ emissions from these targets is key to tracking the correct application and effectiveness of the reduction
policies, and supporting the assessment of the global stocktake implemented by the United Nations. Satellite observations
have the advantages of providing periodical data and having potential global coverage. Furthermore, as initially proposed by
Bovensmann et al. (2010) and Velazco et al. (2011), analysis of space-based observations of $XCO_2$, the column-averaged dry
air mole fraction of $CO_2$, provide independent estimates of $CO_2$ emissions from localized sources like power plants (Reuter
et al., 2019; Nassar et al., 2017, 2022).

The detection of $CO_2$ emission plumes from localized sources is challenging due to the small anomaly in the $XCO_2$ due
to these emissions in the atmosphere, which are typically in the order of one ppm and in the same order of magnitude as the
instrument noise (Bovensmann et al., 2010). Since net atmospheric $CO_2$ has a long lifetime, ranging from years to millennia
(Ciais et al., 2013), and large fluxes of natural origin, these enhancements are also much smaller than $CO_2$ background values
and natural variability.

Nitric oxide (NO) is co-emitted with $CO_2$ during the combustion of fossil fuels. It rapidly reacts with ozone ($O_3$) to form
nitrogen dioxide ($NO_2$). During the day, $NO_2$ is photolyzed to produce NO and atomic oxygen. Therefore, NO and $NO_2$ are
coupled during the daytime and their sum is referred to as $NO_x$. Unlike $CO_2$, $NO_x$ has a lifetime in the order of hours in the
daytime boundary layer. As a result, $NO_2$ vertical column densities in plumes released from fossil fuel combustion exceed
background values and sensor noise typically by orders of magnitude. This makes it a suitable tracer for recently emitted $CO_2$.
The approach of using $NO_2$ as a proxy for recent $CO_2$ emissions for the combustion of fossil fuels has been successfully
used previously, both to estimate the $CO_2$ emissions from $NO_x$-to-$CO_2$ emission ratios (Reuter et al., 2014; Hakkarainen
et al., 2021) and to detect and constrain the spatial extent of the emission plume, using observed data (Reuter et al., 2019) as
well as synthetic observations (Kuhlmann et al., 2019, 2021). The use of $NO_2$ as a proxy for $CO_2$ profits from simultaneous
observations of both gases for an increased correlation in the spatial structures. We investigated this technique using currently
available observations of $XCO_2$ and column densities of $NO_2$, retrieved from the Orbiting Carbon Observatory 3 (OCO-3)
and the TROPOspheric Monitoring Instrument (TROPOMI), respectively. This is in part preparation for the planned extensive





exploitation of this approach to observations from the upcoming Copernicus Anthropogenic $CO_2$ Monitoring (CO2M) mission, which aims to quantify anthropogenic $CO_2$ emissions and will simultaneously retrieve $XCO_2$ and $NO_2$ column densities (Bézy et al., 2019). The CO2M builds on the heritage of the preparatory work undertaken in the CarbonSat concept studies (Buchwitz et al., 2013; Bovensmann et al., 2010) and the observations of the SCIAMACHY on ESA Envisat (Burrows et al., 1995; Bovensmann et al., 1999), Tanso on GOSAT (Kuze et al., 2009, 2016) and OCO (Crisp et al., 2004; Eldering et al., 2019).

Several methods exist to quantify the emissions from localized sources using satellite data, as described e.g. by Varon et al. (2018). The Gaussian plume inversion method, based on the simulation of a Gaussian plume which is then fitted to the observations, has been used to quantify power plant emissions from both OCO-2 and OCO-3 data (Nassar et al., 2022, 2017; Chevallier et al., 2022). The Gaussian model does not, however, account for eddies. We have used a mass-balance cross-sectional flux method on $XCO_2$ retrievals from OCO-3. The cross-sectional flux method, together with the imaging capabilities of OCO-3, allowed us to analyse plume structures. A cross-sectional method was also used by Hakkarainen et al. (2023) to derive $CO_2$ emissions of localized sources in the South African Highveld from OCO-3 data. We focused on the Bełchatów Power Station, in Poland, which is among the power plants having the highest $CO_2$ emissions in the world. This power station was also the object of study by Nassar et al. (2022), who quantified its emissions using OCO-3 data.

This paper is structured as follows. Our cross-sectional flux method for the top-down quantification of the $CO_2$ emissions from localized sources is described in Sec. 2. The datasets used are presented in Sec. 2.1. The plume detection and characterization algorithm, based on TROPOMI $NO_2$ data, is described in Sec. 2.2.1. Section 2.2.2 describes the processing of the $XCO_2$ data to estimate the emission rate as detailed in Sec. 2.2.3. The estimation of the uncertainties is explained in Sec. 2.3. We briefly describe the scene selection procedure (Sec. 2.5) and the method to compute bottom-up emission estimates (Sec. 2.6) to verify the top-down computed emission rates. The results are shown in Sec. 3 and discussed in Sec. 4 along with the conclusions.

## 2   Datasets and methods

### 2.1   Datasets

#### 2.1.1   XCO$_2$

NASA's OCO-3 $XCO_2$ retrievals were the main input data used to derive the $CO_2$ emissions. NASA's OCO-3 instrument, onboard the International Space Station (ISS) since May 2019, measures reflected sunlight in three bands, centred on the molecular oxygen-A band at 0.76 μm and the two $CO_2$ bands at 1.6 μm and 2.0 μm. The instrument has eight footprints, each of 1.6 km, and it sweeps about 2.2 km in the 0.33 seconds integration time. As a result of the ISS precessing orbit, the local overpass time of OCO-3 varies every day and it views latitudes between approximately $\pm 52°$. In its Snapshot Area Map (SAM) mode, it can scan almost adjacent swaths over $CO_2$ emission hotspots and other targets. These SAMs are scans of a region of about 80 km by 80 km, taken in approximately 2 minutes (Eldering et al., 2019; Payne et al., 2022).



We utilized observations taken in SAM mode from the Level 2 Lite $XCO_2$ OCO-3 product (Taylor et al., 2023; O'Dell et al., 2018) in its version 10.4r, based on the Atmospheric Carbon Observations from Space (ACOS) retrieval algorithm (O'Dell et al., 2012; Crisp et al., 2012). These $XCO_2$ estimates are geolocated, bias-corrected and contain a quality flag, which we have used to filter out estimates that are less likely to be accurate. This quality filtering is derived from thresholds on single retrieval variables that are identified to cause the largest differences in the retrieved $XCO_2$ compared to truth proxies (Payne et al., 2022; O'Dell et al., 2018).

### 2.1.2 NO₂

We used TROPOspheric Monitoring Instrument (TROPOMI) $NO_2$ retrievals to detect the shape and location of the $NO_2$ enhancement due to the power plant emission plume. TROPOMI, onboard ESA's Sentinel-5 Precursor (S5P), provides observations on $NO_2$ among other atmospheric constituents (Veefkind et al., 2012). It has a swath of approximately 2600 km across the track of the satellite, divided in 450 ground pixels of about 5.6 km (along track) × 3.6 km (across track) at nadir. It has a nadir-viewing grating spectrometer with four detectors for the different spectral bands: UV and VIS (270-500 nm), NIR (710-770 nm) and SWIR (2314-2382 nm) (Eskes et al., 2022). S5P has a sun-synchronous orbit with a mean local solar time at ascending node of 13:30 h. It performs 14 orbits per day with a repeat cycle of 16 days and its revisit time is approximately one day.

We used slant column densities (SCD) obtained with a Differential Optical Absorption Spectroscopy (DOAS) retrieval in the region from 425 to 497 nm (Richter et al., 2011). The SCD of a trace gas is a measure of its density along the average light path from the Sun to the instrument after reflection at the Earth's surface. Consequently, it depends on the viewing and solar geometry, as well as on other factors like the presence of clouds and aerosols. The vertical column densities (VCD) are related to SCDs through the airmass factor, AMF, as $VCD = \frac{SCD}{AMF}$. The accuracy in the absolute VCDs is less relevant for our application because we do not use them for emission quantification but for detecting enhanced anomalies with respect to background values and thereby quantifying the spatial extent of the emission plume from a localized source. Therefore, we neglected multiple scattering in the atmosphere by the electromagnetic radiation in the spectral region used for the retrieval and approximated the VCDs considering a geometrical AMF from the viewing zenith angle ($\theta_v$) and the solar zenith angle ($\theta_s$) as $AMF = \sec\theta_v + \sec\theta_s$.

### 2.1.3 Meteorological data

We obtained meteorological information from the ERA5 dataset, the fifth generation atmospheric reanalysis of the global climate covering the period from 1940 to present (Hersbach et al., 2017, 2020), produced by the European Centre for Medium Range Weather Forecast (ECMWF) and provided by the Copernicus Climate Change Service (C3S). We used hourly estimates of a number of atmospheric variables in a $0.25° \times 0.25°$ grid and 137 hybrid sigma/pressure vertical levels.

From ERA5 we obtained, for the different vertical layers, the horizontal wind speed components, $u$ and $v$. We also computed the number of dry air molecules in the vertical column from meteorological profiles. Assuming that the emission plume is well-mixed within the boundary layer, we computed, at each location and time, an average of each wind component within the





boundary layer weighted by the number of dry air molecules in the corresponding vertical layer. Brunner et al. (2023) found, with their simulations over the Bełchatów and Jänschwalde power plants in May and June 2018 and in consistency with flight observations, that this assumption of a well-mixed emission plume within the boundary layer is a good approximation during the daytime.

## 2.2 Top-down $CO_2$ emission quantification method

We estimated the net emission rate, $f$, from a localized source using a cross-sectional flux method. This method is based on mass balance so that $f$ is the flux through any cross section (CS) downwind of the source.

Let $\rho$ [kg m$^{-2}$] be a map of the $CO_2$ vertical column mass density at each spatial pixel and $\rho_{bg}$ be the $CO_2$ vertical column mass density map for the background, i.e., the corresponding $\rho$ in the absence of the source under analysis. The anomaly in the

130 vertical column mass density, $\Delta\rho$, resulting from the emissions of the source with emission rate $f$, is given by $\Delta\rho = \rho - \rho_{bg}$ at each spatial pixel. Let $\boldsymbol{w} = (u, v)$ [m s$^{-1}$] be the horizontal wind vector field at plume height and let us consider a CS of infinite length (or whose length is larger or equal to the plume width) through this map and downwind of the source, as sketched in Fig. 1. The normal vector to the CS, $\boldsymbol{n}$, forms an angle $\theta$ with $\boldsymbol{w}$. The mass flux density field is then given by $\boldsymbol{F} = \boldsymbol{w} \, \Delta\rho$. Let us consider as well a positively oriented closed curve C enclosing the emission source (and no other sources), with normal

vector $\boldsymbol{n_C}$ at each point and with one side coincident to the CS. Under stationary conditions, and assuming that all the emitted $CO_2$ mass was transported downwind, the only non-zero flux through this curve is given by the flux through the CS, which is, by mass balance, a measure of the net emission rate:

$$f = \oint_C \boldsymbol{F} \, \boldsymbol{n_C} dl = \int_{-\infty}^{+\infty} \boldsymbol{F} \, \boldsymbol{n} \, \mathrm{d}l = \int_{-\infty}^{+\infty} w_\perp \, \Delta\rho \, \mathrm{d}l \tag{1}$$

where $\mathrm{d}l$ is a length differential along the CS and $w_\perp = w \cos\theta$ the projection of $\boldsymbol{w}$ onto the direction of $\boldsymbol{n}$.

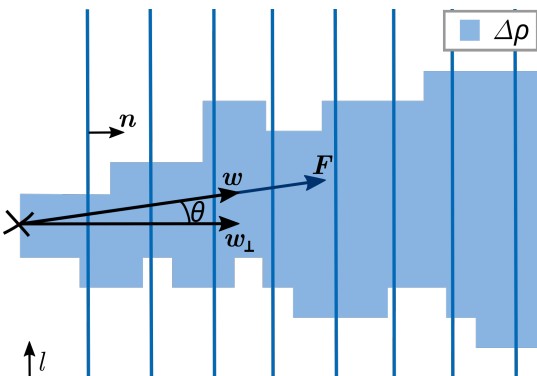

**Figure 1.** Sketch of the cross-sectional flux method. The location of the emission source is marked as a black gross. Several CSs are depicted as blue lines.



The $CO_2$ quantity retrieved by the OCO-3 instrument, $XCO_2$ [ppm], is transformed to vertical column mass density [kg m$^{-2}$] by the formula $\Delta\rho = \frac{M_{CO_2}}{N_A} \Delta XCO_2\, n_d$, where $M_{co_2}$ is the molar mass of $CO_2$ (44.009 g mol$^{-1}$), $N_A$ is the Avogadro number (6.02214076×10$^{23}$ mol$^{-1}$) and $n_d$ is the number of dry air molecules per unit area (estimated from ERA5 meteorological vertical profiles). We then discretized the flux integral in Eq. 1 as the sum over each spatial pixel $i$ along the CS. With this, we rewrote the expression for the cross-sectional flux as:

$$f = \frac{M_{co_2}}{N_A} \sum_i w_{\perp,i}\, \Delta XCO_{2,i}\, n_{d,i}\, \Delta l_i, \qquad (2)$$

where $\Delta l_i$ stands for the length of each spatial pixel along a given CS.

The steps carried out to quantify the emission rate with our cross-sectional flux method are outlined in Fig. 2, where each of the three main blocks is detailed below. The potential plume detection uses TROPOMI $NO_2$ column densities to identify a region containing the emission plume. The $XCO_2$ processing comprises initially the determination of the $XCO_2$ anomaly and a second step that refines the shape of the emission plume. Subsequently, for the emission rate estimation, a set of cross-sectional fluxes is computed to estimate the mean emission rate and its uncertainty.

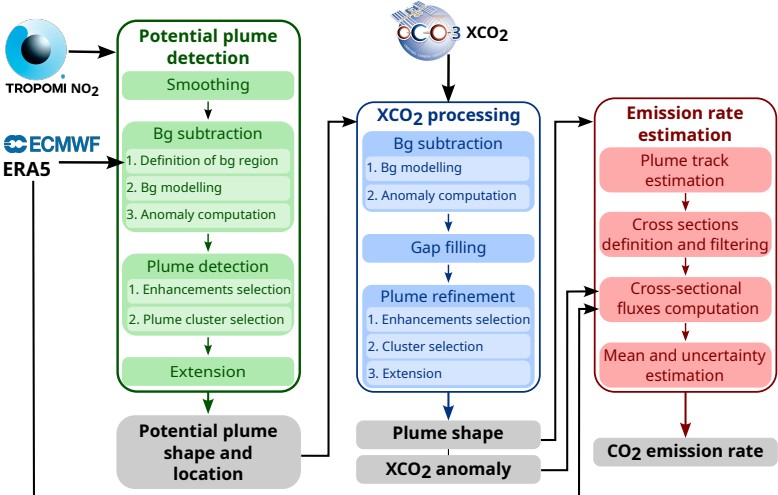

**Figure 2.** Diagram sketching the main steps in the top-down emission quantification algorithm. Each coloured block corresponds to a distinct step, whose output is shown in a gray box at the bottom. The input data points to the steps where they are used. Sub-steps are numbered and shown in boxes within the parent step. For details see Sec. 2.2.1 for step "Potential plume detection", Sec. 2.2.2 for "$XCO_2$ processing" and Sec. 2.2.3 for "Emission rate estimation".

### 2.2.1 Potential $CO_2$ plume detection using $NO_2$ data

The potential $CO_2$ plume detection algorithm essentially defines a region in space that contains the detected $NO_2$ emission plume and is expected to enclose the $CO_2$ emission plume from the source of interest. The algorithm, sketched in the left block in Fig. 2, relies on TROPOMI $NO_2$ VCD (described in Sec. 2.1.2), co-located with the OCO-3 SAM under analysis, and with



a time difference between the S5P and OCO-3 overpasses of less than 5 hours. For the potential plume detection we also need horizontal wind data (Sec. 2.1.3).

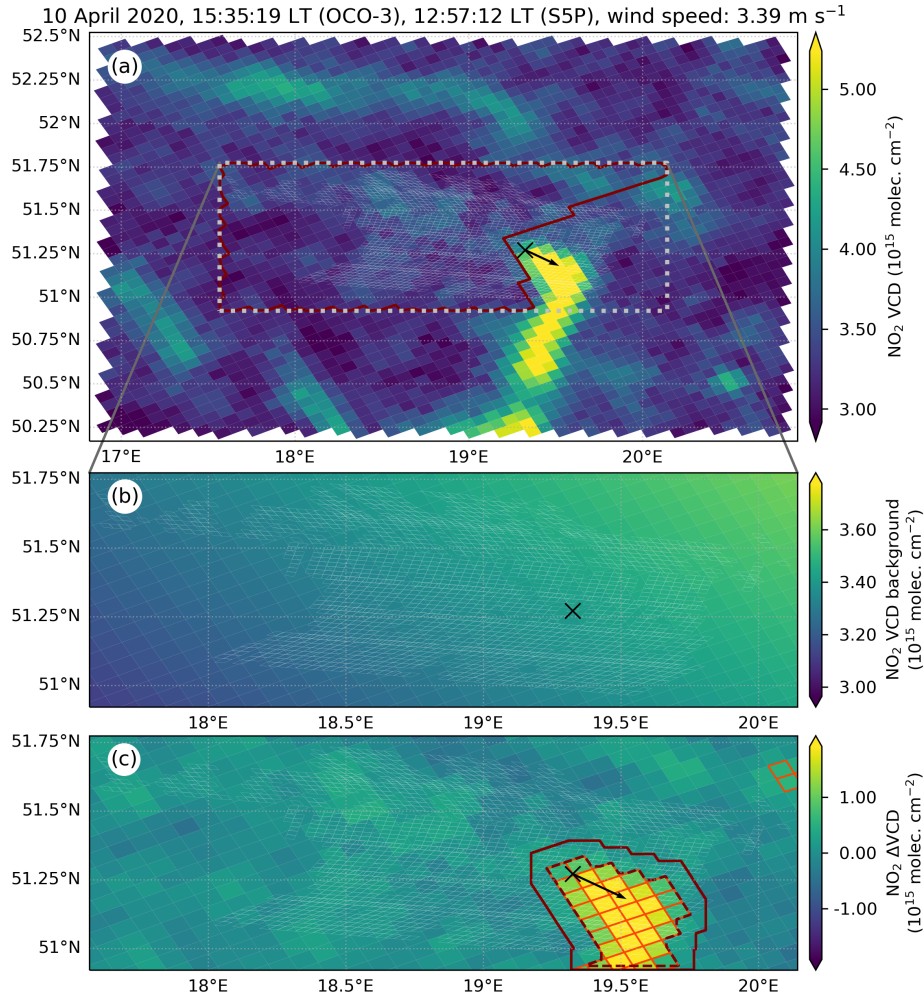

**Figure 3.** Steps of the potential plume detection method using TROPOMI $NO_2$ data for a SAM over the Bełchatów Power Plant on 10 April 2020. The times (in local time) refer to the beginning of the overpass for both OCO-3 and S5P. The location of the source is marked as a black cross. The black arrows show the mean horizontal wind within the potential plume at the OCO-3 overpass time. The borders of the SAM footprints are depicted as white polygons. (a) Smoothed $NO_2$ VCD. The SAM region is depicted as a dashed grey line, and the solid red line encloses the background region. (b) Modelled background. (c) Vertical column density anomaly, $\Delta$VCD. The observations with enhanced $\Delta$VCD as obtained from the significance test are enclosed by orange polygons. The dashed red line surrounds the cluster closest to the source, and the solid red line stands for the potential plume.

The spatial extension of the scene is defined using the OCO-3 SAM. We defined a SAM region as the rectangle enclosing the SAM observations, with a cut-off in longitude and latitude at 2° from the coordinates of the source. This is shown in Fig.





3a as a dashed grey line. We also considered a frame of $0.75°$ around this SAM region. Due to the larger swath of TROPOMI compared to that of OCO-3, the image of $NO_2$ VCD allows us to inspect the surroundings of the $XCO_2$ SAM and identify other potential sources of $CO_2$ around the SAM and thereby exclude sources other than that targeted in the analysis.

We first smoothed the $NO_2$ data to reduce random noise by means of a two-dimensional convolution with a binary kernel. This kernel has the shape of a von Neumann neighbourhood, consisting of the pixel itself and its four first-neighbouring pixels.

In the TROPOMI spatial resolution, this neighbourhood size is often similar to the width of the emission plume close to the source. Its result is essentially the replacement of the VCD in each pixel by the average over a neighbourhood around it, similar to the approach suggested by Kuhlmann et al. (2019) and Varon et al. (2018).

For the background (bg) subtraction, we first defined the $NO_2$ background region, *i.e.* a sector of the scene expected to contain a representative sample of background observations within the SAM area and no signal due to the $NO_2$ emissions.

After taking the averaged horizontal wind speed components at the centre of each TROPOMI pixel within the SAM region at the time of the S5P overpass, we defined a wedge centred along this horizontal wind direction, with its centre slightly displaced upwind of the source, an angular amplitude of $90°$ and a radius long enough to cover the scene. The $NO_2$ background region is the area containing the observations within the SAM region that lie outside this wedge. An example is shown in Fig. 3a as the region enclosed by the solid red line.

Emission plumes reside in the troposphere, while the VCD refers to the whole vertical column. To remove the stratospheric component of the VCD as well as large-scale tropospheric background patterns, we assumed that these VCD components exhibit smooth variations within the scene compared to the portion resulting from anthropogenic emissions from localized sources (Leue et al., 2001). Therefore, to model the background, we fitted the VCD values within the background region to a linear function of longitude and latitude. We subtracted this modelled background from the VCD to obtain the vertical column

anomaly, $\Delta$VCD. Figure 3b shows an example of the modelled background and the corresponding $\Delta$VCD is displayed in Fig. 3c.

We selected the observations with enhanced $\Delta$VCD with respect to the background using a one-tailed Welch test. The null hypothesis is the equality of the background and $\Delta$VCD means. The combined standard error of the mean was computed for each pixel from the background standard deviation and the reported $NO_2$ uncertainty. The observations for which the null

hypothesis was rejected at a significance level, $p$, of 5% were marked as enhancements, enclosed by orange boundaries in Fig. 3c. These enhancements were clustered by Moore neighbourhoods. The $NO_2$ plume is the cluster located closest to the source location, depicted in Fig. 3c as a dashed red line.

The time difference between the S5P and OCO-3 retrievals leads to a decreased correlation in the spatial structures (Lei et al., 2022; Hakkarainen et al., 2023). In that time between overpasses, the atmospheric conditions can vary, which can cause

displacements in the emission plume or alter its shape, disrupting the congruence and overlap between the detected $NO_2$ plume shape and the $CO_2$ plume. This congruence might be further disrupted by the different spatial resolution of OCO-3 and TROPOMI. To obtain a detected potential plume that encloses the $CO_2$ plume, we performed a spatial extension of the $NO_2$ plume mask by binary dilation. For that, we re-gridded the TROPOMI pixels to a high-resolution $0.001° \times 0.001°$ grid. The magnitude of this extension was computed as proportional to the time difference between overpasses, with a minimum of





$0.03°$ in the case of simultaneous overpasses and $0.08°$ for a time difference of 5 hours. This extension increases the likelihood of the $CO_2$ plume being contained within the potential plume. However, the $CO_2$ plume might extend beyond the borders of the potential plume if the wind was highly variable in the time between overpasses. The extended plume, whose boundary is shown in Fig. 3c as a solid red line, is the detected potential plume, which is expected to contain the signal due to the $CO_2$ emissions as well as a fraction of the background observations.

### 2.2.2   $XCO_2$ processing

The processing of the $XCO_2$ to later estimate the emission rate is sketched in the middle block of Fig. 2. We first estimated the $XCO_2$ anomaly, $\Delta XCO_2$, from the quality filtered OCO-3 $XCO_2$ data. We defined a $CO_2$ background region using an extension of the potential plume by binary dilation by about $0.35°$ and excluding the potential plume. This region is depicted in Fig. 4a enclosed by a solid black line and outside the potential plume (solid red contour). We modelled the OCO-3 background
as a fit of the $XCO_2$ observations within the background region to a linear function of longitude, $\lambda$, and latitude, $\phi$. Some SAMs have been observed to present biases between adjacent swaths, likely arising from an interplay between viewing geometry and the presence of aerosols (Bell et al., 2023). This swath bias was accounted for in the linear background model by including an extra term in the equation, $s_j$, for each swath $j = 1, 2, ..., n-1$ where n is the number of swaths of the SAM, so that the $XCO_2$ background model is:

$b_j = a_0 + a_1\lambda + a_2\phi + s_j,$                                                (3)

having a total of $n-1$ equations. An example of a modelled background is shown in Fig. 4b. For each observation, the corresponding background value from the model was subtracted to obtain the $XCO_2$ anomalies, $\Delta XCO_2$, shown in Fig. 4c.

     The horizontal wind components and the number of dry air molecules were obtained, as described in Sec. 2.1.3, at the centre of each OCO-3 footprint within the potential plume at the time of the overpass. The resulting averaged wind vector is
shown in Figs. 3 and 4 as a black arrow. We re-gridded the $\Delta XCO_2$ values and the meteorological information to the same high-resolution $0.001° \times 0.001°$ grid as for $NO_2$ and filled in the missing $\Delta XCO_2$ footprints in the grid using inverse squared distance weighting interpolation for observations within a region of radius $0.05°$ centred at each missing footprint.

     This method relies on the spatial correlation of the $NO_2$ and $CO_2$ emission plumes, which is typically not perfect mainly due to changes in the meteorological conditions, and consequently also in the plume shape and location, in the time between
the S5P and OCO-3 overpasses. The plume extension performed as the last step of the potential plume detection takes into consideration possible mismatches between the detected $NO_2$ plume and the $CO_2$ plume due to these changes at the expense of a larger potential plume, which includes more background observations. This is, in theory, not critical since the $\Delta XCO_2$ within the potential plume should contain only the signal due to the emission plume and random noise that averages out to zero. However, in cases where the background has small-scale structures that have not been characterized by our background model,
the background observations within the potential plume might not average to zero, adding thus a bias. To minimize this bias, we performed a refinement of the potential plume following a similar approach to the plume detection described in Sec.2.2.1. We first masked the footprints with enhanced $\Delta XCO_2$ values with respect to the background by means of a one-tailed z-test





with a p-value of 5%. After a binary closing operation that merges any enhancements separated by less than about 0.06° to obtain a coherent mask, we clustered the enhancements. These clusters are shown in Fig. 4c within dashed orange boundaries.

Any isolated cluster of about the size of a OCO-3 footprint or less (disregarding any filled data) was neglected since it is most likely to be the result of random noise in the $\Delta XCO_2$. With a second binary closing operation, we merged clusters separated by less than about 0.14°. This second closing operation provides us with a coherent mask even if there are relatively large blocks of missing observations within the potential plume. We selected the cluster closest to the source and extended it through binary dilation in 0.015° (about the shortest side of a OCO-3 footprint), obtaining like this the refined plume, shown in Fig. 4c

enclosed by a solid orange line.

### 2.2.3 Emission rate estimation

From the refined plume shape and the $XCO_2$ anomalies, we estimated the mean $CO_2$ emission rate following the steps outlined in the rightmost block in Fig. 2.

We first transformed the high-resolution grid to the local tangent plane (LTP) at the location of the source considering the

Earth geometry to be a WGS84 ellipsoid. For the footprints within the refined plume, we carried out a linear regression of their coordinates to define the plume track, shown in Fig. 4c as a red straight line traversing the refined plume and passing through the source coordinates. Along this track and perpendicular to it, we defined a number N of equidistant CSs separated by a distance, $\Delta x$, of approximately 0.2 km. This track, together with its perpendicular CSs on the LTP, spans a new coordinate system, hereinafter referred to as *track coordinate system*, whose resolution in the x-axis (along the plume track) is determined

by the distance between consecutive CSs and in the y-axis is set to about 0.1 km along the given CS. The transformation between the high-resolution grid and the track coordinate system comprises a rotation of the coordinate axes followed by an undersampling procedure, where only the footprints that are crossed by a CS are taken into account. This undersampling with respect to the high-resolution grid does not lead to a loss of information because the resolution of the track coordinate system is still about one order of magnitude higher than the original OCO-3 resolution. In this transformation, the refined plume mask

was slightly modified insofar that, if the mask has any hole along a CS, it is filled. The resulting $\Delta XCO_2$ data transformed to this coordinate system is shown in Fig. 5a.

We accepted only the subset of CS at distances downwind of the source larger than 5 km and smaller than 35 km. We refer to the span in between as the plume range. The lower threshold avoids errors due to both the OCO-3 product geolocation error, typically of less than 1 km and reaching up to 3 km for a fraction of the data (Payne et al., 2022), and due to the exact location

of the stacks, which can be about 1 km apart. Furthermore, it accounts for the fact that the assumption of good vertical mixing of the plume within the boundary layer is realistic only after about the height of the boundary layer downwind of the source, typically in the order of 1-2 km (Matheou and Bowman, 2016). For larger distances downwind of the source, diffusion causes the dilution of the plume, reducing the $XCO_2$ enhancement, which can lead to the detection of only a fraction of the plume extension, and therefore an underestimation of the computed cross-sectional fluxes. This was avoided with the upper limit of

the plume range.




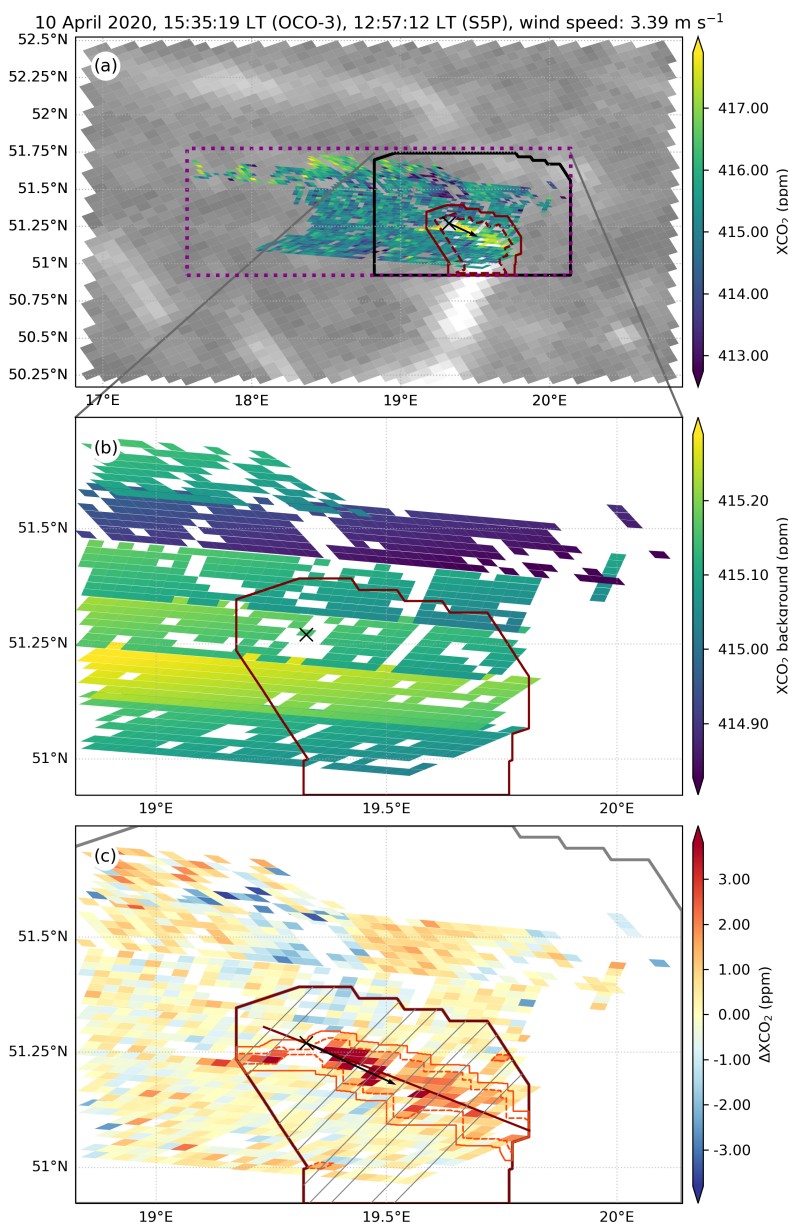

**Figure 4.** Steps for $XCO_2$ processing for the flux computation for a OCO-3 SAM over the Bełchatów Power Plant, on 10 April 2020. The location of the source is marked as a black cross. The black arrows stand for the average wind within the potential plume at OCO-3 overpass time. The potential plume is enclosed by a solid red line. (a) $XCO_2$ footprints (colour-coded) over $NO_2$ VDC in the background (gray scale). The dashed purple line delimits the SAM region. The $XCO_2$ background area is enclosed by the black line and outside the potential plume. The dashed red line encircles the $NO_2$ detected plume. (b) Modelled $XCO_2$ background according to Eq. 3. (c) $\Delta XCO_2$. The refined plume is enclosed by the solid orange line within the potential plume. Other clusters with enhanced $\Delta XCO_2$ are enclosed by dashed orange boundaries. A number of valid CS along the track is shown as grey lines. The straight red line that traverses the potential plume is the computed plume track.



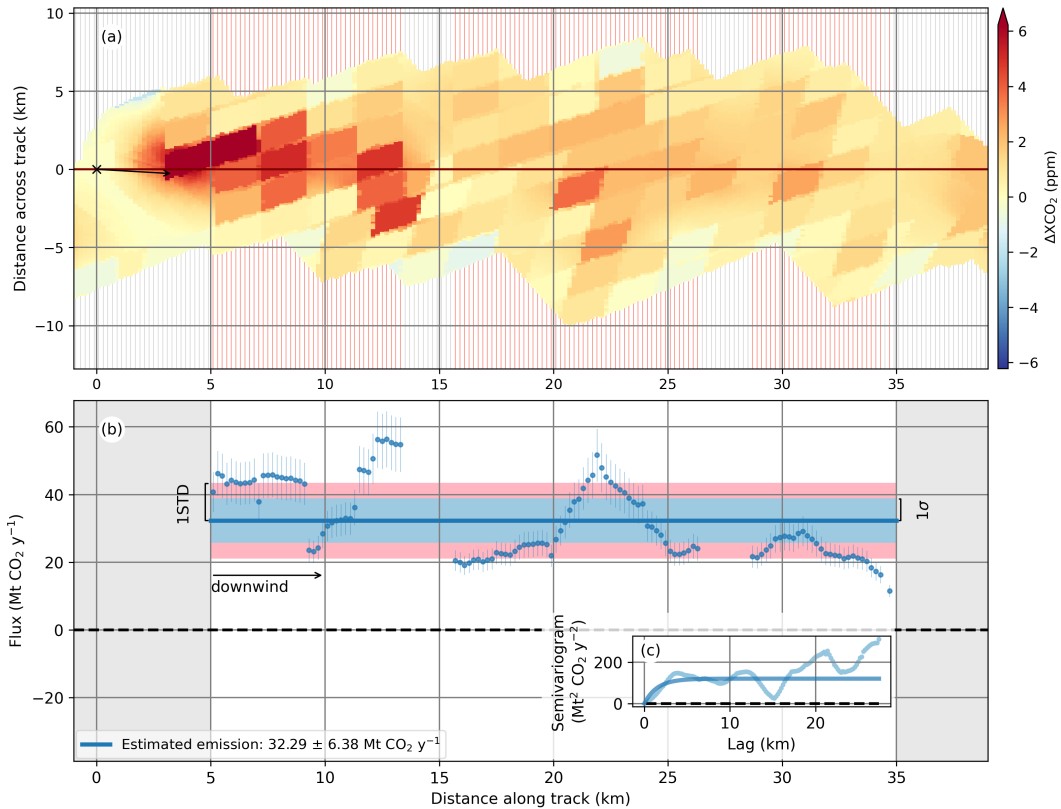

**Figure 5.** Example of the emission rate estimation procedure, as explained in Sec. 2.2.3, for the scene on 10 April 2020. The x-axis represents the distance of the CSs from the source (at the origin) along the plume track. (a) $\Delta XCO_2$ within the refined plume in the track coordinate system. The black arrow stands for the averaged horizontal wind within the potential plume. The red lines stand for valid CSs and the grey lines for invalid CSs. The filled $\Delta XCO_2$ gaps are shown together with the observations. (b) Estimated fluxes across each valid CS along the plume track. The blue dots account for the cross-sectional fluxes used for the estimation of the mean emission rate, shown as a solid blue line. The standard deviation of the data is shown as a pink area to both sides of the mean line, and the $1\sigma$ error, calculated as described in Sec. 2.3, appears as a light-blue area. The vertical bars at each dot account for the propagation uncertainty (see Sec. 2.3.2). The fluxes through valid CSs at distances from the source outside the plume range are plotted as blue crosses. (c) Semivariogram used to compute the dispersion uncertainty (see Sec. 2.3.1). The dots stand for the empirical semivariogram, computed using Eq. 5. The model resulting from the exponential fit (Eq. 6) is depicted as a solid line.

In addition, we filtered out CSs that are likely to yield biased estimates of the emission rate. We considered the CSs to be valid if they fulfilled all of the following conditions: a) less than 40% of the $XCO_2$ observations within the refined plume along the CS are missing, to avoid considering CSs with too much interpolated data added to fill gaps, b) the width of the refined plume along the CS is larger than 4 km, which ensures that the CS spans over more than one SAM footprint. After filtering out
CSs outside the plume range and applying conditions a) and b), we are left with a subset of $n'$ CSs.





With this, we computed, using Eq. 2, a set of values $\{f_j\}$, for $j = 1, 2, ..., N$, corresponding to the $CO_2$ flux through each CS at distance, $x_j$ from the source along the plume track, where only a subset of $n'$ valid CSs was considered. An example of these cross-sectional fluxes is shown in Fig. 5b as scattered blue dots along the plume track. We expect the obtained cross-sectional fluxes to vary along the track of the plume. These fluctuations are partly originated by the experimental error in the quantities
used in Eq. 2, but also due to the turbulent nature of the process.

In addition, the $CO_2$ molecules observed in the SAM were released at different times. The farther away the CS is from the source, the longer the $CO_2$ molecules were released before the OCO-3 overpass time. Let us define the *plume characteristic time*, $\Delta t$, as the time that the $CO_2$ molecules would have needed to travel, at the mean horizontal wind speed within the potential plume, the distance between the source and the valid CS within the plume range situated the farthest away from the
source. If the power plant emissions vary within this $\Delta t$, they will add another source of fluctuations to the cross-sectional fluxes. We calculated this plume characteristic time and rounded it to the nearest hour integer. For a typical plume length along its track of about $30 \text{ km}$, the characteristic time ranges from approximately 1 to 3 hours, for wind speeds between 3 and 7 $\text{ms}^{-1}$.

To describe the process leading to the flux fluctuations we took a stochastic approach. Let each $f_j$ be a realization of a
random variable $F(x_j)$, at points $x_j$, $j = 1, 2, ..., N$ along the plume track. To characterize $F$, we assume second-order or weak stationarity (WS), which means that: a) the mean of $F$ is constant for all $x_j$, which allows us to estimate the mean of the process by treating the $f_j$ at different locations as realizations of the same random variable. This assumption is realistic as long as both the emissions from the power plant and the wind vector have no significant trend within the characteristic time interval. b) The covariance, $\text{Cov}(F(x_j), F(x_{j+\delta}))$ of the random variables at points $x_j$ and $x_{j+\delta}$, $\forall \delta = 0, 1, 2, ..., N-1$ depends only
on the distance between two points along the plume track, $d = |x_{j+\delta} - x_j|$, but not on their absolute positions. Since all the $x_j$ are equally spaced, the distance between any pair of consecutive locations will be constant and given by $\Delta x = x_{j+1} - x_j$. Therefore, we can write the lag distance as $d = \delta \Delta x$, where $\delta$ is the *lag index*, and consequently $\text{Cov}(F(x_j), F(x_{j+\delta})) = C(\delta)$, where $C$ is the covariance function. On account of the first assumption, we can estimate the mean emission rate, $\bar{f}$, from the mean of the computed cross-sectional fluxes. The obtained results are detailed in Sec. 3.2 and Table 1. With the second
assumption, we can as well estimate its uncertainty.

## 2.3 Uncertainty

To estimate the uncertainty of the obtained mean emission rate, we considered three contributions: 1) The *dispersion uncertainty*, $s_{\text{disp}}$, includes all random effects that cause cross-sectional fluxes to oscillate about their mean. These effects have different origins, such as the inherent variability of the cross-sectional fluxes due to turbulence, variations in the emissions
within the plume characteristic time or random errors in the quantities used to compute the cross-sectional fluxes. 2) The *wind uncertainty*, $s_{\text{wind}}$, refers to the impact on the emission rate estimate of a possible bias in the horizontal wind speed. 3) The *sensitivity uncertainty*, $s_{\text{sens}}$ includes the effect on the emission estimate of different choices of the analysis parameters. Considering all three sources of uncertainty are uncorrelated, the standard error ($1\sigma$) of the mean emission rate, $s(\bar{f})$, is therefore





given by $s^2(\bar{f}) = s^2_{\text{disp}} + s^2_{\text{wind}} + s^2_{\text{sens}}$. Each of these contributions to the uncertainty are explained below and the corresponding

results shown in Table 1.

### 2.3.1   Dispersion uncertainty

The dispersion uncertainty is given by the variance of the mean:

$$s^2_{\text{disp}} = Var(\bar{f}) = \frac{1}{n^2} \sum_{j=1}^{n} \sum_{k=1}^{n} Cov(F(x_j), F(x_k)) \underset{\text{WS}}{=} \frac{1}{n} \left( C(0) + 2 \sum_{\delta=1}^{n-1} (1 - \frac{\delta}{n}) C(\delta) \right), \tag{4}$$

where the last term is the explicit sum over the elements of the covariance matrix for a weakly stationary process divided

by $n^2$ (Storch and Zwiers, 1999). Thus, we can compute the dispersion uncertainty provided knowledge of the shape of the covariance function.

For uncorrelated data, $C(\delta) = 0$ for $\delta \geq 1$, so Eq. 4 reduces to $s^2_{\text{disp}} = \frac{1}{n} C(0)$. However, the cross-sectional fluxes are spatially correlated, especially due to the close spacing between consecutive CSs compared to the OCO-3 spatial resolution. Since the effective number of independent CSs is unknown, we performed a correlation analysis to estimate the covariance function,

$C(\delta)$, which we used to estimate the dispersion uncertainty using Eq. 4.

We used a semivariogram, defined as $\gamma(x_j - x_k) = \frac{1}{2} Var[F(x_j) - F(x_k)]$, to estimate the covariance function. For a weakly-stationary process, it can be written as $\gamma(\delta) = \frac{1}{2} Var[F(x_{j+\delta}) - F(x_j)]$ and fulfils that $\gamma(\delta) = C(0) - C(\delta)$, showing, in this case, the equivalence between the semivariogram and the covariogram. When estimated from the data, the semivariogram is preferred because it does not require the knowledge of the mean and is, therefore, an unbiased estimator (Montero et al., 2015).

We can empirically estimate the semivariogram as:

$$\hat{\gamma}(\delta) = \frac{1}{2m(\delta)} \sum_{j=1}^{m(\delta)} (f_{j+\delta} - f_j)^2, \tag{5}$$

where $m(\delta)$ is the number of pairs of data, $\{f_{j+\delta}, f_j\}$ used for the estimation. For a pair to be taken into account, both $f_{j+\delta}$ and $f_j$ must correspond to valid CSs. Therefore, $m$ decreases for larger lags, $\delta$, and depends on the number and distribution of valid CSs.

The empirical semivariogram is only defined for a subset of the total $N$ lags and it might exhibit an erratic behaviour. A widely used solution is to use a model to fit the estimated semivariogram. A suitable model should represent some basic features, like a monotonic increase with increasing lag, which shows decreasing correlation until the sill (horizontal asymptote) is reached, and a positive intercept that accounts for the *nugget effect*, *i.e.* a discontinuity at the origin (Webster and Oliver, 2007). We use an exponential model of the shape:

$$\gamma(\delta) = C(0)(1 - e^{-\delta \Delta x / l}), \tag{6}$$

where $C(0)$ is the variance of the process. The parameter $l$ is a measure of the correlation length and the only free parameter to be determined by the fit. This model assumes that there is no nugget effect since we expect a smooth variation of the semivariances due to the close spacing between CSs. Since the estimation of the empirical semivariances is less reliable for





larger lag distances and smaller number of pairs, $m$, used for the computation, we considered only the empirical semivariances
computed from a number of pairs $m \geq 10$.

With the modelled semivariogram, we computed the covariance function for each lag, $\delta$, as $C(\delta) = C(0) - \gamma(\delta)$, which we used to compute the dispersion uncertainty making use of Eq. 4.

The effective number of independent CSs within the plume range was estimated as:

$$n'_{\text{eff}} = \frac{C(0)}{s^2_{\text{disp}}} \tag{7}$$

For a typical plume range of 30 km and about 2 km footprint width we would expect a maximum of about 15 independent CSs. Further correlation among CSs derived from spatial structures would lead to $n'_{\text{eff}} \leq 15$. $n'$ provides us, therefore, with an intuitive check on the computed dispersion uncertainty. It is worth noting that $n'_{\text{eff}}$ is independent of the number, $n'$, of valid CSs within the plume range, provided $n'$ was large enough to perform the correlation analysis.

### 2.3.2   Wind uncertainty

The wind uncertainty, $s_{\text{wind}}$, includes the effect on the estimated emission rate of a possible bias in the horizontal wind used to compute the cross-sectional fluxes. This is a purely systematic component, since any fluctuations in the wind along the plume track are accounted for in the dispersion uncertainty.

We considered an uncertainty on the horizontal wind speed perpendicular to the CSs, of $s(w_\perp) = 0.5 \ \text{ms}^{-1}$. This value is representative of the ERA5 ensemble spread zonal wind component close to the surface (Hersbach et al., 2020). Using Eq. 2
and substituting $w_\perp$ by its uncertainty, we obtain a value of the effect of the wind uncertainty on each cross-sectional flux, $b_{w,j}$. These uncertainties $b_{w,j}$ are shown in Fig. 5b as blue bars about the flux estimates. The wind uncertainty is the mean effect of the wind bias on the emission rate estimate, *i.e.* the mean of $b_{w,j}$.

### 2.3.3   Uncertainty from sensitivity

The sensitivity uncertainty, $s_{\text{sens}}$ is a measure of the effect on the emission estimate of different choices of the parameters used
for the analysis.

We estimated a measure of this uncertainty contribution from the variation of a number of parameters used for the analysis of each scene within plausible ranges: 1) the p-values for the detection of the potential plume and plume refinement, in both cases ranging from 0.03 to 0.1, 2) the radius of the circle and power of the inverse distance to fill the SAM gaps within the refined plume. We varied the radius from $0.05°$ to $0.1°$ for inverse distance weighting and from $0.05°$ to $0.2$ from inverse
squared distance weighting. 3) The limits of the plume range, for which we varied the lower limit from 3 to 10 km and the upper limit from 30 to 40 km from the source. 4) The function used to fit the background, where we considered three cases: linear dependence on longitude and latitude with a possible swath bias (given by Eq. 3), an analogous model allowing as well for a possible footprint bias, $f_k$ for $k = 1, 2, ..., 7$, and a model considering only the linear dependence on longitude and latitude (*i.e.* setting $s_j = 0 \ \forall j$ in Eq. 3).




For each parameter, we computed the standard deviation of the emission estimates for each scene and took its mean as a measure of the sensitivity uncertainty for that parameter. Assuming uncorrelated errors, we added them quadratically to compute an estimate of $s_{\text{sens}}$.

## 2.4 Sensitivity tests

Additionally to the sensitivity analysis performed to obtain an uncertainty estimate, we have performed other sensitivity tests.
These tests are explained in what follows, and the results are presented in Sec. 3.3. The modifications evaluated in these tests have shown to either have no significant influence on the results or lead to biased emission estimates. Therefore, we have not included the outcome of these tests in the sensitivity uncertainty.

a) In the computation of the cross-sectional fluxes, the wind speed and direction from ERA5 were used. It is a common practice to rotate the wind vector to match the direction of the observed plume (Reuter et al., 2019; Nassar et al., 2017; Varon
et al., 2018). We have therefore tested the influence of the wind rotation to match the direction of the detected plume track.

b) The OCO-3 L2 $XCO_2$ product includes a quality flag, allowing us to filter out those observations tagged as having poorer quality. Omitting the quality filtering has the advantage of a denser coverage, thus having fewer missing SAM observations at the expense of higher bias risk.

c) We used $NO_2$ VCD to obtain the shape of the potential plume, which constrains the $CO_2$ plume region. We tested the
applicability of the method without using $NO_2$ measurements but constraining the $CO_2$ plume region by a wedge downwind of the source (as described in Sec. 2.2.1) to define the $NO_2$ region.

## 2.5 Scene selection

The analysed scenes were selected using an automatic procedure. It comprises searching for SAMs with more than 800 soundings (after quality filtering) and co-located TROPOMI $NO_2$ overpasses with a time difference to the OCO-3 overpass smaller
than 5 hours.

We also applied additional filters in the emission quantification procedure. A scene was discarded according to the following criteria: 1) If there were less than 20 OCO-3 observations within the detected potential plume or less than 50 in the background. This condition discards scenes with too few observations for our emission estimation procedure. 2) The width of TROPOMI pixels increases in the across-flight direction due to the increasing viewing angle, from about 3.5 km at nadir to about 14
km at the edges. A larger pixel size can result in an underestimation of the extension of the detected $NO_2$ plume with the statistical test due to the dilution of the signal, further intensified due to smoothing. In cases with strong $NO_2$ emissions, the larger pixel size can also lead to an overestimation of the detected plume, underconstraining the $CO_2$ plume and leading to a possible inclusion of background structures in the $CO_2$ plume. To avoid such cases, we left out a scene if more than 50% of the TROPOMI observations with enhanced VCD belonged to the outer 50 pixels of the swath (on any side). 3) If the angle
between the plume track and the wind direction was wider than 45°. This provides an additional check on the wind direction as obtained from ERA5 and avoiding the analysis of scenes where there was an abrupt change in the wind direction shortly before the overpass. 4) If the highest computed lag distance was smaller than 2 km (about the size of an OCO-3 footprint). With this





criterion we avoid characterizing the dispersion of the fluxes along the plume track with an insufficient number of independent CSs.

## 2.6 Bottom-up emission estimation

To validate our top-down emission estimates, we computed the $CO_2$ emissions of the Bełchatów Power Station for the selected scenes at approximately the OCO-3 overpass time using a bottom-up approach based on the power plant activity.

First, an hourly bottom-up estimate of the $CO_2$ emissions was computed as the product of the hourly generated power and the emission intensity (mass of emitted $CO_2$ per unit of generated power). We used information on the hourly net generated power per generation unit of 100 MW or more installed capacity, provided by the European Network of Transmission System Operators for Electricity (ENTSO-E) in its Transparency Platform (https://transparency.entsoe.eu/, last accessed on 3 Feb 2023). The emission intensity was computed from the total $CO_2$ emissions divided by the net generated power by the power plant in 2018. That year, the $CO_2$ emissions were 38.4 Mt $CO_2$, as reported by the European Industrial Emissions Portal (https://industry.eea.europa.eu, last accessed on 20 Feb 2023). The European Industrial Emissions Portal collects information from the EU Registry on Industrial Sites and the European Pollutant Release and Transfer Register (E-PRTR). The net generated power in that year, according to the power plant operator (PGE), was 32.535 TWh. That yields a $CO_2$ intensity of 1.18 $10^{-6}$ Mt $CO_2 (MWh)^{-1}$, which we assumed to have remained approximately constant up to 2022.

Power plant emissions have strong daily and day-to-day variations (Velazco et al., 2011). For this reason, despite providing our top-down emission estimates in units of $MtCO_2 \, y^{-1}$, they are not annual averages. They are up-scaled and represent the emissions within approximately the plume characteristic time, $\Delta t$, before the OCO-3 overpass. Therefore, we averaged the reported hourly generated power within that $\Delta t$ before the OCO-3 overpass to compute the bottom-up emission estimates, which we can directly compare with our top-down estimates.

We estimated the uncertainty of the bottom-up emission estimates by taking into account two contributions: the *intensity uncertainty* and the *characteristic time uncertainty*. The former refers to the uncertainty on the estimated emission intensity, subject to the uncertainty of the data used in its computation. Gurney et al. (2016) analysed two emission datasets for power plants in the U.S., finding monthly emission differences of about 6% for about half of the facilities. For such a large power plant, we would not expect the uncertainty to belong to the top 50 percentile. Despite that, we considered a conservative 6% uncertainty on the $CO_2$ emissions from EIEP (2020), and consequently on the emission intensity computed from it. We assumed that this includes changes in the intensity over the years, the different emission intensities that the various units in the power plant have and possible mismatches between the net generated power as reported by ENTSO-E and the power plant operator. We also considered the *characteristic time uncertainty*. This contribution to the uncertainty refers to the mentioned emission variations within $\Delta t$, which we account for by taking half the maximum difference in the hourly generated power times the emission intensity. The total uncertainty of our bottom-up estimates is the root sum of squares of the characteristic time and the intensity uncertainties.

Our bottom-up emission estimates are shown in Table 1, together with their corresponding uncertainties.





## 3 Results

### 3.1 Scene selection

In the period from the beginning of the OCO-3 $XCO_2$ dataset, in July 2019, until November 2022 inclusive, we found a total of 94 SAMs over the Bełchatów Power Plant, of which 14 have more than 800 soundings (after quality filtering). All these 14

SAMs have at least one co-located S5P overpass with a time difference smaller than 5 hours. After applying the scene selection filters mentioned in Sec 2.5, we were left with 9 scenes, corresponding to 9 different SAMs.

### 3.2 CO$_2$ emission estimates

The results for these 9 scenes obtained from the scene selection are detailed below, illustrated in Fig. 9 and summarized in Table 1 along with the meteorological information for each scene and several other parameters that characterize the scene. The

correlation between our top-down (TD) and bottom-up (BU) estimates is 0.92, and the results obtained by these two methods agree in 8 out of 9 cases within their $1\sigma$ uncertainty range. Their mean difference (TD - BU) is -2.8 $MtCO_2 \, y^{-1}$ and their standard deviation is 3.7 $MtCO_2 \, y^{-1}$. The mean uncertainty for these 9 scenes is 5.8 $MtCO_2 \, y^{-1}$ (22.0%) and it is dominated by the dispersion uncertainty, which is on average about 1.8 times higher than the wind uncertainty and slightly larger (on average 1.3 times higher) than the sensitivity uncertainty.

The results obtained for the 9 analysed scenes are detailed below. They are illustrated in Figs. 3-8 for some of the overpasses, and in Figs. A1-A5 in Appendix A for those overpasses not shown in this text. We refer to overpass times in local time (LT) as determined by the corresponding time zone. All the overpass times for the analysed scenes for the Bełchatów power station refer to Central European Summer Time (CEST).

**10 April 2020.** The OCO-3 overpass began at 15:35 LT (CEST) and the co-located S5P overpass at 12:57 LT. Our top-down

$CO_2$ emissions are estimated to be 32.29 $\pm$ 6.38 $MtCO_2 \, y^{-1}$. The mean horizontal wind speed within the potential plume was 3.39 $m \, s^{-1}$ and had an angle relative to the plume track of about -4.7°. The characteristic time was estimated to be of about three hours. Within that characteristic time before the OCO-3 overpass, all the units were operative and the bottom-up estimated emissions decreased by 3.51 $MtCO_2 \, y^{-1}$. Within that time there was a gradual increase in the wind speed by 1.5 $m \, s^{-1}$ and an angle shift of 13.94° about the plume track. The steps to compute the emission estimate

and the results are shown in Figs. 3-5.

**17 April 2020.** The OCO-3 overpass was at 11:42 LT, and the S5P overpass began at 14:06 LT. Our top-down emission estimate is 32.04 $\pm$ 10.27 $MtCO_2 \, y^{-1}$, illustrated in Fig. 6. The averaged wind speed within the potential plume at the OCO-3 overpass time was 5.86 $m \, s^{-1}$, having an angle of 18.4° with the detected plume track. We estimated a characteristic time of approximately 2 hours. Within that characteristic time before the OCO-3 overpass, the wind speed

slightly increased to then decrease by a total amount of about 0.79 $m \, s^{-1}$, and its angle remained approximately constant. We refer to an approximately constant wind speed or wind direction if its change within the characteristic time is less than 0.5 $m \, s^{-1}$ and 10°, respectively. The power plant activity slightly decreased within the characteristic time, from 3381 to



3041 MWh, leading to a bottom-up age uncertainty of $1.63 \, \mathrm{MtCO_2 \, y^{-1}}$. The dispersion uncertainty of $9.28 \, \mathrm{MtCO_2 \, y^{-1}}$ is the largest among all the analysed scenes. In Fig. 6c,d we can appreciate the oscillations in the cross-sectional fluxes, with a $CO_2$ accumulation at higher distances from the source.

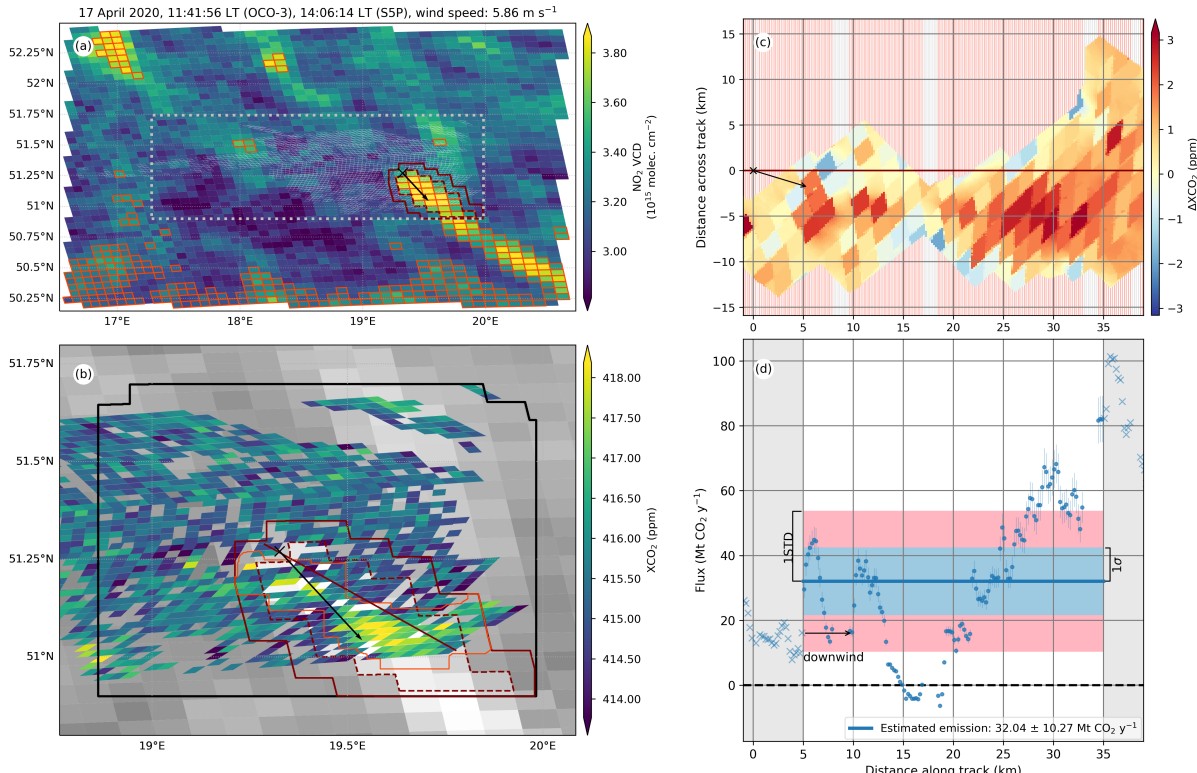

**Figure 6.** Overview of the top-down emission rate estimation steps for the scene on 17 April 2020. (a) $NO_2$ VCD. The SAM footprints are enclosed by grey polygons. As in Fig. 3, the observations with enhanced $\Delta$VCD as obtained from the t-test are enclosed by orange polygons. The SAM region is depicted as a dashed grey line. The dashed red line surrounds the cluster closest to the source, and the solid red line accounts for the potential plume. (b) $XCO_2$ over $NO_2$ VDC in the background in a gray scale. The solid black line encloses the $XCO_2$ background area (excluding the potential plume). The refined plume is enclosed by the solid orange line within the potential plume. The straight line that traverses the potential plume is the computed track. (c) Analogous to Fig. 5a. (d) Analogous to Fig 5b. The black arrows in a-c depict the mean horizontal wind within the potential plume. The black cross stands for the source location.

**18 June 2021.** Figure 7 shows the results for this overpass. The top-down emission estimate is $32.77 \pm 5.40 \, \mathrm{MtCO_2 \, y^{-1}}$. Within the 2 hours characteristic time prior to the overpass, the wind speed increased by $0.71 \, \mathrm{m \, s^{-1}}$ and its direction remained approximately constant. So did the generated power, as shown in the relatively small age uncertainty ($0.12 \, \mathrm{MtCO_2 \, y^{-1}}$).







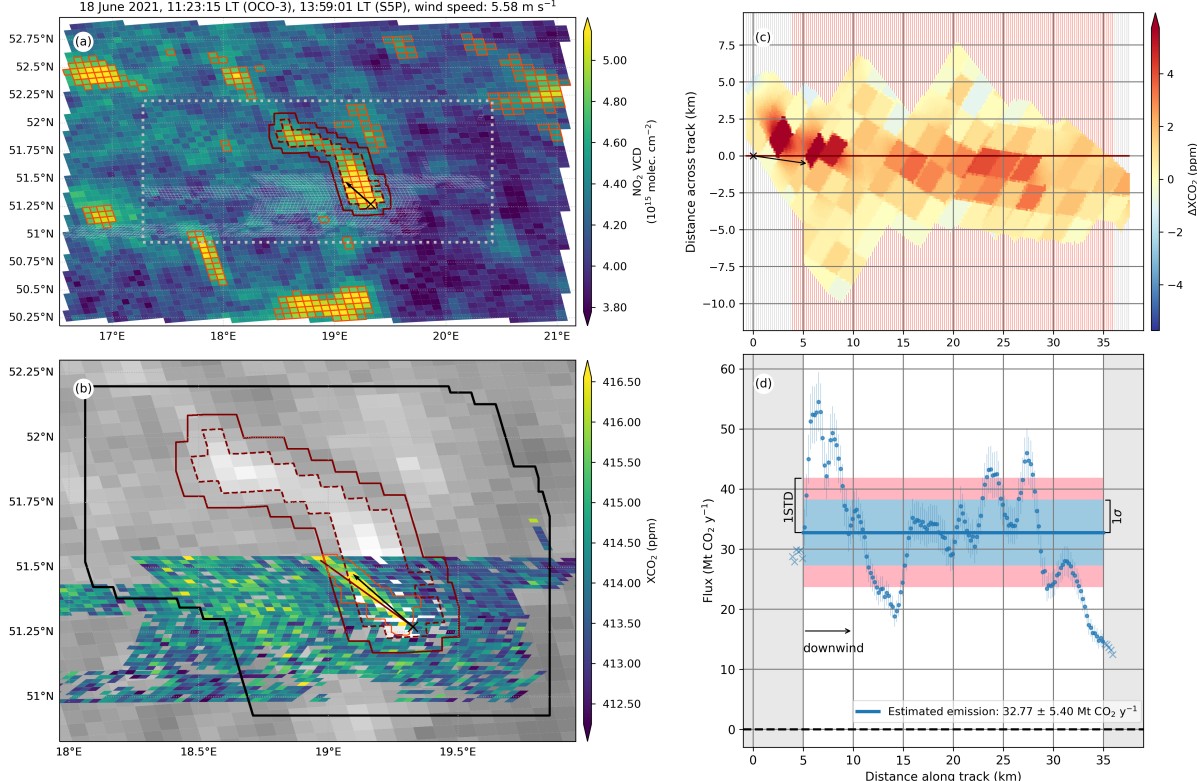

**Figure 7.** Overview of the top-down emission rate estimation steps for the scene on 18 June 2021. Analogous to Fig. 6.

**19 June 2021.** The potential plume extends, in this case, to areas without OCO-3 observations. Therefore, we have obtained cross-sectional fluxes only up to a distance from the source of about 23 km. Our top-down emission estimate is $41.94 \pm 6.99 \ \mathrm{MtCO_2 \ y^{-1}}$. The results for this overpass are shown in Fig. A1.

**20 June 2021.** The low power plant activity, operating at less than 40% of its maximum capacity, together with the relatively large wind speed of $6.80 \ \mathrm{m \ s^{-1}}$, results in enhancements over the background of the same order of magnitude as the background structures, so that the emission plume is hardly perceptible at first sight (see Fig. A2). The estimated top-down emissions are $17.96 \pm 3.83 \ \mathrm{MtCO_2 \ y^{-1}}$. Within the one hour (characteristic time) prior to the OCO-3 overpass, the wind speed decreased by about $0.61 \ \mathrm{m \ s^{-1}}$, and its direction and the power plant generated power remained approximately constant. The large fraction of missing OCO-3 observations within the refined plume led to a discard of about two-thirds of the defined CSs within the plume range, especially at distances between 14 and 23 km from the source. The low number of valid CSs, distributed in blocks spanning over less than 5 km and with gaps reaching 10 km, led to a likely incomplete characterization of the correlation of the cross-sectional fluxes and thus to an underestimation of the dispersion uncertainty ($1.88 \ \mathrm{MtCO_2 \ y^{-1}}$). The inference of an underestimated dispersion uncertainty can also be reached when looking at the unexpectedly high computed effective number of independent CSs.



**8 October 2021.** The large fraction of missing (quality filtered) SAM observations led to the near absence of valid CS to
compute the fluxes for distances smaller than about 23 km downwind of the source. We observed (Fig. A3) a near
monotonic increase in the fluxes with the distance downwind to the source over about 10 km. This increase could be
attributed, at the first instance, to a violation of the stationarity assumption that we made to estimate the mean emission
rate and its uncertainty. However, the change in the generated power within the characteristic time is in the order of 10%,
while the fluctuations in the cross-sectional fluxes are at least one order of magnitude larger. The wind speed and the
power plant activity have remained approximately constant within the characteristic time of one hour before the overpass.
Therefore, a plausible cause of the apparent monotonic change in the cross-sectional fluxes is the turbulent flow, where
we missed part of the oscillating behaviour due to the cut-off of 35 km along the plume track. This explanation seems
consistent with Fig. A3d, where we observe a decrease in the cross-sectional fluxes at distances greater than 35 km
downwind.

**9 October 2021.** The results are shown in Fig. 8. The wind speed and power generation are comparable to those determined
for the scene on 20 June 2021. The relatively large fraction of gaps within the potential plume and low enhanced $\Delta XCO_2$
led to an apparent underestimation of the refined plume extension before about 18 km downwind of the source. This
is the only case where the obtained top-down estimate ($11.59 \pm 4.12$ $MtCO_2$ $y^{-1}$) does not agree with our bottom-up
estimate ($19.54 \pm 1.48$ $MtCO_2$ $y^{-1}$) within the uncertainty ranges.

**24 June 2022.** The OCO-3 overpass took place at 9:01 am and is the earliest of all the scenes investigated. Despite the 4-hour
time difference with the TROPOMI overpass, which is the largest of the analysed scenes, the detected potential plume
seems to contain well the OCO-3 plume (see Fig. A4). The wind was highly variable: in the characteristic time of 1 hour
before the OCO-3 overpass the wind speed increased by 3.73 $m\,s^{-1}$ and its direction changed by 13.35°.

**13 October 2022.** The top-down emission estimate is $27.75 \pm 6.08$ $MtCO_2$ $y^{-1}$. Within the 1-hour characteristic time before
the overpass, the wind speed and direction remained constant, and there was a drop in the generated power, from 3045
MW to 2464 MW, resulting in a higher age uncertainty than for other scenes, but smaller than the dispersion uncertainty.
This overpass is illustrated in Fig. A5.





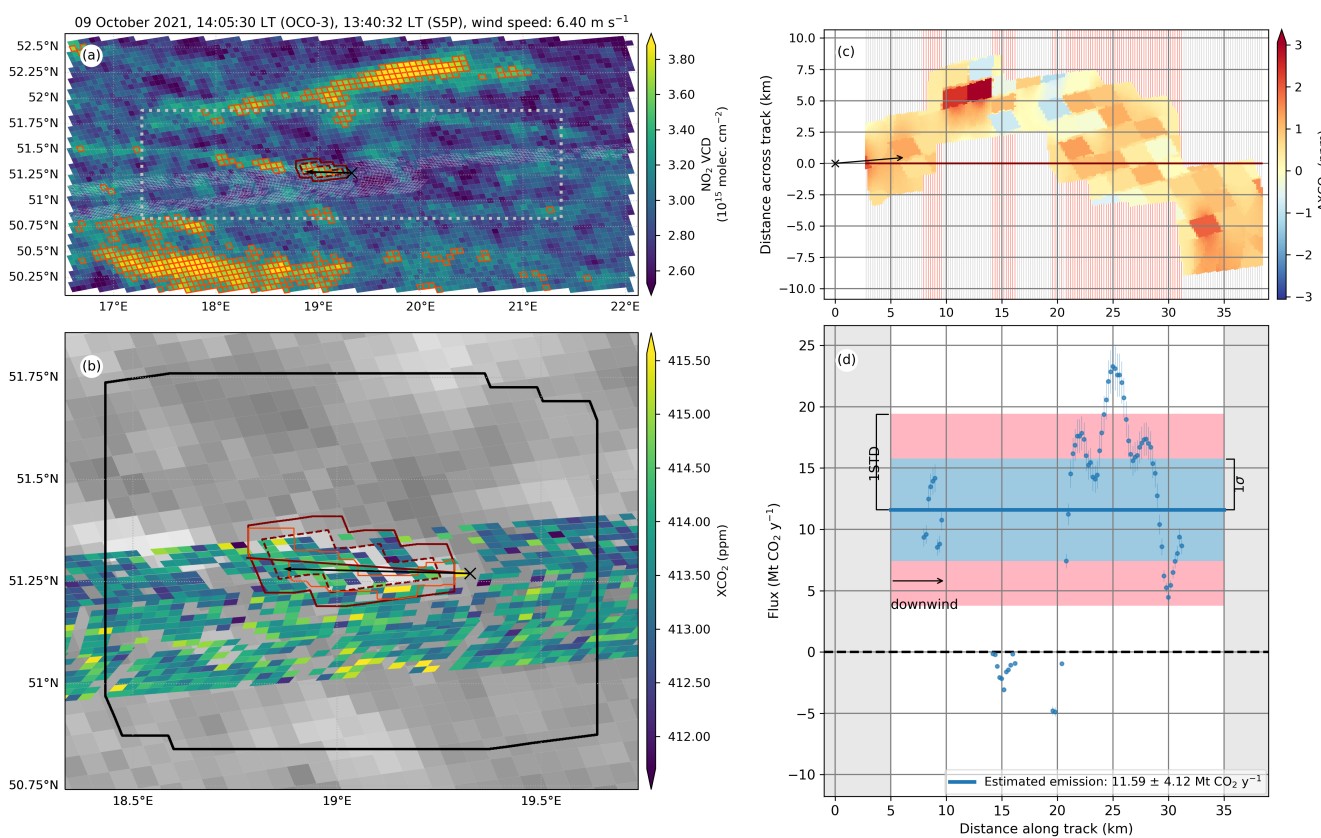

**Figure 8.** Overview of the top-down emission rate estimation steps for the scene on 9 October 2021. Analogous to Fig. 6.





**Table 1.** Parameters characterizing each of the analysed scenes including $CO_2$ emission estimates. The satellites overpass times at local time are shown, as well as the meteorological information at OCO-3 overpass time. The angle refers to that between the wind vector and the north-to-south direction, positive in the clockwise direction. The top-down and bottom-up emission estimates are shown together with their corresponding uncertainties, broken down into their components as described in Sec. 2.3. The sensitivity uncertainty of 3.11 Mt $CO_2$ y$^{-1}$ is included in the total top-down uncertainty estimates.

| | Date | 2020 10 Apr | 2020 17 Apr | 2021 18 Jun | 2021 19 Jun | 2021 20 Jun | 2021 08 Oct | 2021 09 Oct | 2022 24 Jun | 2022 13 Oct |
|---|---|---|---|---|---|---|---|---|---|---|
| | OCO-3 time (LT) | 15:35 | 11:42 | 11:23 | 10:36 | 09:48 | 14:53 | 14:06 | 09:01 | 12:34 |
| | S5P time (LT) | 12:57 | 14:06 | 13:59 | 13:40 | 13:21 | 14:00 | 13:41 | 13:03 | 13:22 |
| | Wind speed (m s$^{-1}$) | 3.39 | 5.86 | 5.58 | 6.18 | 6.80 | 7.78 | 6.40 | 9.10 | 5.76 |
| | Angle $\theta$ (°) | -53.53 | -30.35 | 144.60 | 139.92 | 139.06 | 93.84 | 92.39 | 131.18 | 151.37 |
| | $n_d$ ($10^{25}$ cm$^{-2}$) | 2.12 | 2.10 | 2.11 | 2.11 | 2.10 | 2.15 | 2.15 | 2.11 | 2.12 |
| | Boundary layer height (km) | 1.64 | 1.32 | 1.26 | 1.26 | 1.03 | 0.79 | 1.06 | 1.23 | 0.65 |
| | Characteristic time (h) | 3 | 2 | 2 | 1 | 1 | 1 | 1 | 1 | 1 |
| | Number of independent CS, $n'_{eff}$ | 10.38 | 5.41 | 7.30 | 6.64 | 24.90 | 8.37 | 9.31 | 15.56 | 5.98 |
| | Generated power (MW) | 3774 | 3217 | 3709 | 3714 | 1936 | 2793 | 1889 | 3208 | 2754 |
| Top-down[a] | Emissions | 32.29 | 32.04 | 32.77 | 41.94 | 17.96 | 24.13 | 11.59 | 33.75 | 27.75 |
| | Total uncertainty | 6.38 | 10.27 | 5.40 | 6.99 | 3.83 | 5.04 | 4.12 | 4.40 | 6.08 |
| | Dispersion uncertainty | 3.40 | 9.28 | 3.33 | 5.37 | 1.88 | 3.61 | 2.55 | 2.49 | 4.62 |
| | Wind uncertainty | 4.42 | 3.11 | 2.89 | 3.22 | 1.21 | 1.65 | 0.90 | 1.86 | 2.45 |
| | Emissions (Nassar et al., 2022) | 29.60 | 27.90 | 36.90 | 26.40 | 10.20 | 21.40 | 16.40 | 35.20 | - |
| | Uncertainty (Nassar et al., 2022) | 3.30 | 1.90 | 4.60 | 1.10 | 1.50 | 3.80 | 2.60 | 7.00 | - |
| Bottom-up[a] | Emissions | 39.04 | 33.28 | 38.37 | 38.42 | 20.03 | 28.90 | 19.54 | 33.19 | 28.49 |
| | Total uncertainty | 2.79 | 2.46 | 2.14 | 2.16 | 1.16 | 1.61 | 1.48 | 1.98 | 3.20 |
| | Age uncertainty | 1.76 | 1.63 | 0.12 | 0.34 | 0.34 | 0.08 | 1.01 | 0.72 | 2.78 |
| | Intensity uncertainty | 2.17 | 1.85 | 2.13 | 2.14 | 1.11 | 1.61 | 1.09 | 1.84 | 1.58 |
| | Emissions (Nassar et al., 2022) | 31.30 | 28.10 | 32.20 | 24.40 | 17.20 | 24.30 | 15.50 | 29.50 | - |
| | Uncertainty (Nassar et al., 2022) | 1.57 | 1.41 | 1.61 | 1.22 | 0.86 | 1.22 | 0.78 | 1.48 | - |

[a]All quantities expressed in Mt $CO_2$ y$^{-1}$.





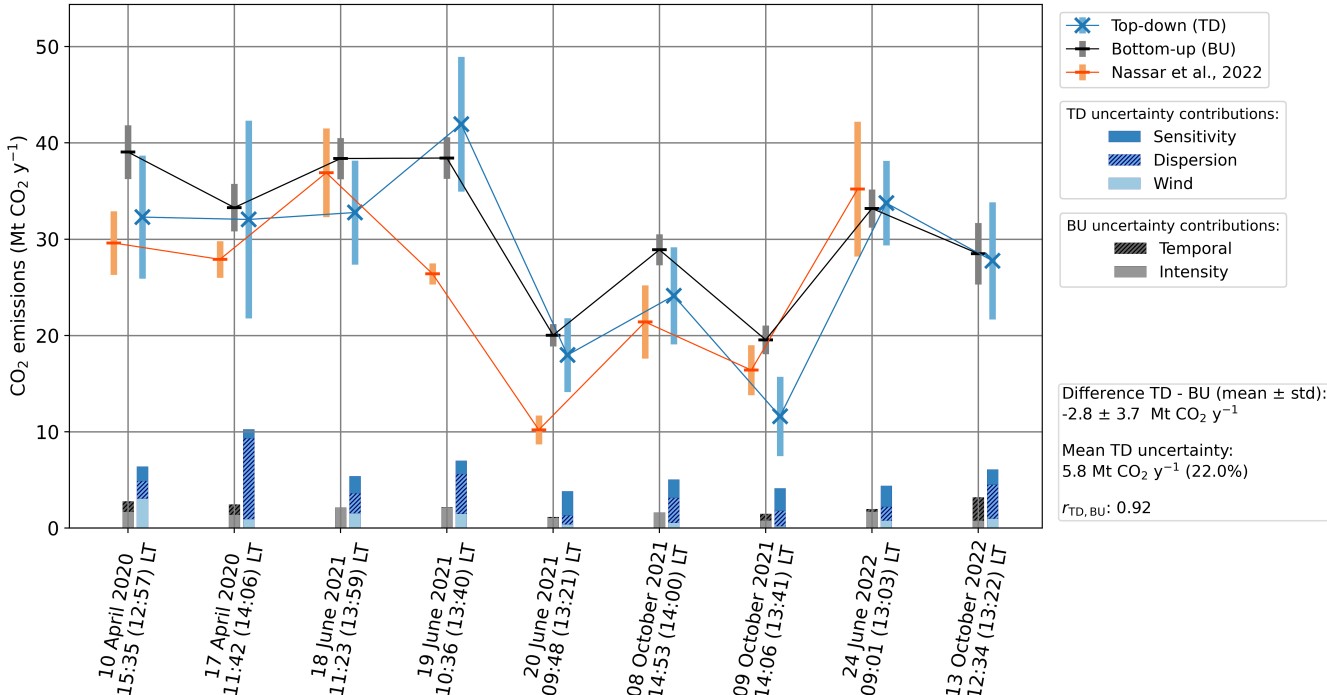

**Figure 9.** Bottom-up (black), top-down (blue) and Nassar et al. (2022) (orange) emission estimates for the analysed scenes. The $1\sigma$ uncertainties are displayed as bars about the corresponding emission estimate. The same uncertainties are shown at the bottom, revealing the relative contributions to the bottom-up and top-down emission estimates, where the bars length is the respective uncertainty contribution quadratically scaled with respect to the total uncertainty.

### 3.3 Sensitivity analysis

As a result of the sensitivity analysis explained in Sec. 2.3.3, we obtained a measure of the sensitivity uncertainty, which
is included in the total uncertainties of our top-down estimates. We obtained sensitivity uncertainties of 1) 1.24 and 1.36 $MtCO_2\,y^{-1}$ for the p-value sensitivity for the potential plume detection and plume refinement, respectively. 2) For the filling parameter sensitivity, we obtained a value of 0.74 $MtCO_2\,y^{-1}$. 3) The lower and upper limit of the plume range led to uncertainties of 0.70 and 1.24 $MtCO_2\,y^{-1}$, respectively. 4) The background model resulted in a sensitivity uncertainty of 1.94 $MtCO_2\,y^{-1}$. Assuming uncorrelated uncertainty contributions, this results in a total $s_{sens}$ of 3.11 $MtCO_2\,y^{-1}$.
In the sensitivity tests a-c described in Sec. 2.4 we analyzed a) wind rotation to match the detected plume track, b) omission of quality filtering of $XCO_2$ data and c) omission of the use of $NO_2$ data to detect the potential plume. The results that we obtained for these tests are summarized below and shown in Fig. A7 in the supplementary material (Appendix A).

a) When automatically rotating the wind direction to match that of the detected plume track, we did not observe significant differences in the obtained emission rates (see Fig. A7a) because the angle that the mean wind speed forms with the detected





plume track was, for the analysed scenes, between $1.4°$ and $18.4°$. The absolute mean difference between bottom-up and top-down slightly decreased (from -2.8 to -2.4 $MtCO_2\ y^{-1}$) and its standard deviation increased in 0.3 $MtCO_2\ y^{-1}$.

b) A larger disparity was found when switching off the quality filtering in the $XCO_2$ data. In this case, summarized in Fig. A7b, we found that running the same analysis including the $XCO_2$ observations considered to have a poor quality results in a correlation coefficient of about 0.45, and a standard deviation of the difference bottom-up minus top-down of 14.7 $MtCO_2\ y^{-1}$.

This discrepancy was specially remarkable in the scenes where the emission plume is close to the lignite pit, situated just a few kilometres south-west of the Bełchatów power plant, where a region of elevated $\Delta XCO_2$ is noticeable in most of the SAMs. In these scenes, the observations with elevated $\Delta XCO_2$ were masked as belonging to the plume. If we discard the scenes in which the wind blows towards the pit region (about $90°$), *i.e.* the scenes on 8 and 9 October 2021, we obtain a correlation coefficient of 0.86 and the difference TD-BU becomes $-3.50 \pm 5.91$ $MtCO_2\ y^{-1}$. Two additional scenes passed our scene selection filters

in this case: on 27 June 2022 and on 10 October 2022.

c) Omitting the use of $NO_2$ data to detect the potential plume led as well to a noticeable decrease in the correlation coefficient (to 0.26) and a TD - BU difference of $-2.2 \pm 10.1$ $MtCO_2\ y^{-1}$, as we can see in Fig. A7c. The main reason for the decreased correlation is the larger potential plume, which underconstrained the $CO_2$ plume region, resulting in the inclusion in the detected plume of neighbouring background structures of enhanced $\Delta XCO_2$, e.g. on the SAMs on 18 June 2021 (shown in

Fig. A8) and 24 June 2022. In addition, no $CO_2$ plume was detected for the SAM on 20 June 2021 using the same p-value as with the use of $NO_2$ data. A higher p-value leads to the detection of the $CO_2$ plume in this case, but also to a further inclusion of background structures in the detected plume. A higher sensitivity (2.77 $MtCO_2\ y^{-1}$) to the chosen p-value was found.

## 4  Discussion and conclusions

With our data-driven cross-sectional flux method using co-located $CO_2$ and $NO_2$ satellite data, we were able to quantify the

$CO_2$ emissions from the Bełchatów Power Station. We estimated the power plant $CO_2$ emissions for 9 automatically identified different OCO-3 overpasses and compared the results with bottom-up (BU) emission estimates, finding a good correlation (0.92). The results obtained by these two methods agree in 8 out of 9 analysed cases within their uncertainty range.

Nassar et al. (2022) have also analyzed 8 of our 9 scenes. Their results are shown in Fig. 9 and Table 1 along with our results. We analyzed an additional scene on 13 October 2022, not shown by Nassar et al. (2022). On the other hand, Nassar

et al. (2022) have shown a SAM corresponding to the 27 June 2022, which was discarded by our filtering algorithm (Sec. 2.5) due to the lack of plume observations left after dumping the OCO-3 retrievals flagged as being of poor quality. Our top-down (TD) estimates agree with those obtained from Nassar et al. (2022) in 6 out of 8 cases. The emission estimates of Nassar et al. (2022) (NA) have a correlation coefficient of 0.85 with our BU estimates, and the difference NA - BU (mean $\pm$ standard deviation) is $-5.8 \pm 4.8$ Mt $CO_2\ y^{-1}$. The correlation coefficient between NA and TD is 0.76, with a mean difference NA - TD

(mean $\pm$ standard deviation) of $-2.8 \pm 6.7$ Mt $CO_2\ y^{-1}$. Nassar et al. (2022) have also computed bottom-up emission estimates based on the generated power by the power plant. However, their bottom-up estimates are scaled by their mean top-down emission estimates. Therefore we have not performed any comparisons with the bottom-up estimates of Nassar et al. (2022).





The relative uncertainties for individual overpasses lie between 13% and 32% (22.0% on average), higher than those obtained by Nassar et al. (2022), which are 3.8-19.7% (12.2% on average). The obtained relative uncertainties are in the same order of magnitude as the uncertainty levels to be achieved with the CarbonSat mission (Buchwitz et al., 2013; Bovensmann et al., 2010), which aimed for about 20% uncertainty on the $CO_2$ emission estimate for individual overpasses (ESA, 2015). The dispersion uncertainty dominates over that from wind and sensitivity, because it accounts for the large fluctuations in the cross-sectional fluxes. Brunner et al. (2023) showed with simulated plumes that estimated individual (2-km-wide) cross-sectional fluxes fluctuate about 20-30% due to turbulence, even with perfect knowledge of the $\Delta XCO_2$ map and wind speed. A similar outcome was obtained by Wolff et al. (2021). We have observed more pronounced fluctuations with standard deviations ranging from 27% to 67% of the corresponding mean emission estimate. We expect these larger fluctuations to arise from the use of modelled data in the mentioned studies, as opposed to our measurement-based analysis. Despite the large fluctuations in individual cross-sectional fluxes, having multiple CS downwind of the source enabled their correlation to be investigated, which lead to dispersion uncertainties between about 1.88 and 9.28 $MtCO_2\,y^{-1}$. To obtain a qualitative check on the obtained dispersion uncertainties, we computed the effective number of CSs for each scene using Eq. 7, as shown in Table 1. In agreement with the reasoning made in Sec. 2.3.1, we obtained typical effective numbers of about 15 CSs or less. A noticeable exception is the unexpectedly high effective number of CSs (24.90) obtained for the scene on 20 June 2021 (Fig. A2). This is probably a consequence of the low number of valid CSs, distributed in blocks of about 5 km or less, with gaps between the blocks reaching 10 km, which likely led to an incomplete characterization of the correlation of the cross-sectional fluxes and in this case to an underestimation of the dispersion uncertainty.

The sensitivity uncertainty (3.11 $MtCO_2\,y^{-1}$) shows a fair stability of the method over the used parameters. All the contributions to the sensitivity uncertainty accounted for are in the same order of magnitude. The choice of the p-value was of little influence for most of the analysed scenes, for both the plume detection and the refinement, as long as it was large enough to detect the full extension of the actual emission plume and there were no other structures with elevated $\Delta XCO_2$ close to the plume. The choice of the p-value only had a significant effect (about 5-6 $MtCO_2\,y^{-1}$) for the plume detection in the scene on 20 June 2021 (Fig. A2), due to the structures with elevated $\Delta XCO_2$ in the vicinity of the plume, included within the potential and refined plume for higher p-values. We encountered a similar situation when setting the upper limit of the plume range along its track, with very small fluctuations for every scene but that on 17 April 2020 (Fig. 6), where the estimated emission rate increased about 10 $MtCO_2\,y^{-1}$ when varying the parameter from 30 to 40 km. This results from an accumulation of $CO_2$ at those distances along the plume track.

In some of the analysed scenes there seemed to be deviations from our assumption of stationarity. For example, we observed significant wind speed variability within the characteristic time for the overpass on 24 June 2022, and less notable on 10 April 2020, 17 April 2020 or 18 June 2021. We also observed noticeable changes in the power plant generated power, as occurred on 13 October 2022. These deviations are partly considered in the dispersion uncertainty because they can enhance or reduce the fluctuations and be partly masked by them. For example, a monotonic decrease in the wind speed would lead to an underestimation of the cross-sectional fluxes, the more pronounced the farther away from the source. For a relatively large decrease by 1 $ms^{-1}$ in the wind speed, from a typical wind speed of about 6 $ms^{-1}$ and emissions of 30 $MtCO_2\,y^{-1}$, variations





of less than about 5 $\mathrm{MtCO_2\,y^{-1}}$ in individual CSs would be expected, which are much smaller than the oscillations observed in the cross-sectional fluxes.

We have identified no significant difference between considering the wind direction obtained from ERA5 and rotating it to match the detected plume track. For this reason, we have used the ERA5 wind speed, because this has the advantage being able to account for variable wind directions along the plume track. In addition, it is independent of any assumption on the shape of the plume track (in our case linear) and any possible difference between our detected plume track and the plume centreline. An advantage of the wind rotation would be a potential increase in the number of analysed scenes, because we have discarded

scenes with an angle between the detected track and the mean wind direction larger than $45°$ (see Sec. 2.5). However, such a large difference between the detected track and the mean wind direction may indicate a bias in the wind vector, and therefore discarding that scene is yet the most prudent option.

We have observed significant disagreement between bottom-up and top-down estimates when using the $XCO_2$ observations flagged as having poor quality. Nevertheless, after discarding the scenes where the emission plume was close to the lignite

pit, the disagreement between performing or not performing quality filtering was significantly smaller. The better agreement between top-down and bottom-up estimates when disregarding these scenes suggests the presence of possible artefacts in the $XCO_2$ estimates over the pit regions, as well as the importance for the application of our method of accurate $XCO_2$ measurements and a reliable quality filtering in case of potential biases in the $XCO_2$. Yet, an analysis with more scenes is needed for a more conclusive result. A larger number of scenes were successfully analysed using our method, at the cost of a

reduced correlation between bottom-up and top-down estimates.

The difference TD-BU had a large standard deviation when the potential plume was not detected using $NO_2$ data, but through a wedge centred at the mean wind vector. This disagreement appears to result from the inclusion in the detected plume of background structures with enhanced $\Delta XCO_2$ (in about the same order of magnitude as the $\Delta XCO_2$ resulting from the power plant emissions) close to the source. The reason for this is that the potential plume defined through this wedge

downwind has a greater extension than that detected using $NO_2$, and therefore it constrains less the region for the $CO_2$ plume detection. Some examples of this inclusion of background structures in the refined plume happened for the overpasses on 18 June 2021 (shown in Fig. A8) and 24 June 2022. In these cases, the discrimination between background features and signal due to the source emissions is difficult without ancillary data. In addition, no $CO_2$ plume was detected for the SAM on 20 June 2021 without the aid of $NO_2$ data. With this, we confirmed the usefulness of $NO_2$ data to constrain the $CO_2$ plume region. It

helped us define a $CO_2$ background region and exclude false positives, *i.e.* pixels wrongly assigned to belong to the plume.

The presented method has some limitations. It can only quantify $CO_2$ emissions of isolated sources. The use of $NO_2$ allows us to identify scenes and targets whose emission plumes might be affected by other emission sources, but no de-coupling has been attempted. In addition, the method relies on the confinement of the $CO_2$ plume within the detected potential plume. In general, we found good agreement. However, we might encounter cases, such as the scene on 18 June 2021 (Fig. 7), where the

$CO_2$ plume seems to extend beyond the potential plume boundaries. In this case, the part of the $CO_2$ plume that we miss is beyond the plume range, having no effect on the final result, but it could lead to an underestimation of the emissions in other cases. This mismatch is due to a change in the wind direction in the time between OCO-3 and S5P overpasses and will be



solved with the use of simultaneously retrieved $XCO_2$ and $NO_2$ maps (at the same spatial resolution) from the future CO2M, from which we expect to detect potential plumes with higher congruity with the $CO_2$ one.

Furthermore, we have focused on one of the power plants with the highest emissions in the world. We have obtained successful top-down estimates for individual overpasses with estimated bottom-up emissions as low as about 19-20 $MtCO_2 \ y^{-1}$. This suggests the feasibility of tracking power plants whose emissions are about that magnitude. Power plants emitting over 20 $MtCO_2 \ y^{-1}$ are responsible for roughly 5% of the total annual power plant $CO_2$ emissions (Strandgren et al., 2020). The uncertainty of the presented method is expected to scale with the source emissions. The dispersion uncertainty includes

terms that are expected to be proportional to the emissions, e.g. those resulting from turbulence effects, as well as other terms that are independent of the emissions, such as those derived from $XCO_2$ random error. The sensitivity uncertainty also has terms proportional to the emissions, resulting e.g. from the uncertainty derived from the gap filling method, and terms that are independent of the emissions, such as those resulting from the uncertainty derived from the function used to fit the $XCO_2$ background. The wind uncertainty is proportional to $\Delta XCO_2$, so we can assume that this component of the total uncertainty

will also be directly proportional to the emissions. Therefore, we would expect uncertainties with a similar proportionality factor as obtained in the present study (22% of the emission rate) for power plants whose emissions are comparable to those of Bełchatów. For power plants with lower $CO_2$ emissions, the absolute total uncertainty is expected to decrease accordingly, with a lower threshold determined by the terms independent of the emissions. These terms will presumably lead to a higher relative uncertainty for power plants with lower $CO_2$ emissions.

With our cross-sectional flux method, we have shown the potential to monitor $CO_2$ emissions from individual power plants using OCO-3 $XCO_2$ observations. With such a method, we can obtain independent emission estimates, which are crucial for facilities with limited or missing information on the activity data. The TROPOMI $NO_2$ column densities have proved useful to detect the emission plume in scenes with other neighbouring sources or small-scale background structures with enhanced $XCO_2$. The application of our method to observations from the planned CO2M is expected to have many advantages. CO2M

simultaneous measurements of $NO_2$ and $XCO_2$, at the same spatial resolution and similar resolution as that of OCO-3, will increase the spatial correlation between $NO_2$ and $XCO_2$ images, and thus allow us to constrain the $CO_2$ plume better, which will lead to an increased accuracy of the emission estimates and a reduced uncertainty.

*Data availability.* The OCO-3 $XCO_2$ is available at https://doi.org/10.5067/970BCC4DHH24, last accessed on 31 January 2023. Generated power per generation unit from ENTSO-E is available at https://transparency.entsoe.eu/, last accessed on 3 February 2023. The ERA5

meteorological dataset is available at the Copernicus Climate Change Service (C3S) Climate Data Store (CDS), last accessed on 28 February 2023.

*Author contributions.* MR, MB and BFA designed the analysis and interpreted the results. BFA developed the code and wrote the manuscript with contributions from all co-authors. AR produced the $NO_2$ data. HBov, HBoe, AR and JPB contributed to improve the manuscript.



*Competing interests.* Some authors are members of the editorial board of *Atmospheric Measurement Techniques*.

*Acknowledgements.* Financial support was provided by the German Meteorological Service, DWD, grant number 4819EMF06A, under the RiGHGorous project. The OCO-3 $XCO_2$ data were produced by the OCO-3 project at the Jet Propulsion Laboratory, California Institute of Technology, and obtained from the OCO-3 data archive maintained at the NASA Goddard Earth Science Data and Information Services Center. The TROPOMI $NO_2$ data were produced by the Institute of Environmental Physics, University of Bremen. ERA5 meteorological profiles have been obtained from the Copernicus Climate Change Service (C3S) operated by the ECMWF. This publication contains modified
Copernicus Sentinel-5 Precursor data. We also thank C. Gerbig (MPI-BGC, Jena) for helpful comments and inputs in an early stage of this study. Part of this work is funded by the BMBF project "Integrated Greenhouse Gas Monitoring System for Germany - Observations (ITMS B)" under grant number 01 LK2103A.

## Appendix A: Additional figures

In Sec. 3 in the main text of this manuscript we described each of the analysed scenes and detailed the results obtained for each
of them. Some of these scenes were also displayed in Figs. 3-8. Figures A1-A5 illustrate the obtained results for the scenes mentioned in Sec. 3 but not shown there.

The sensitivity analysis performed to obtain the uncertainty from sensitivity was described in Sec. 2.3.3. The obtained results are shown in Sec. 3.3 and also illustrated in A6, which shows the sensitivity analysis to the considered parameters, as described in Sec. 2.3.3.




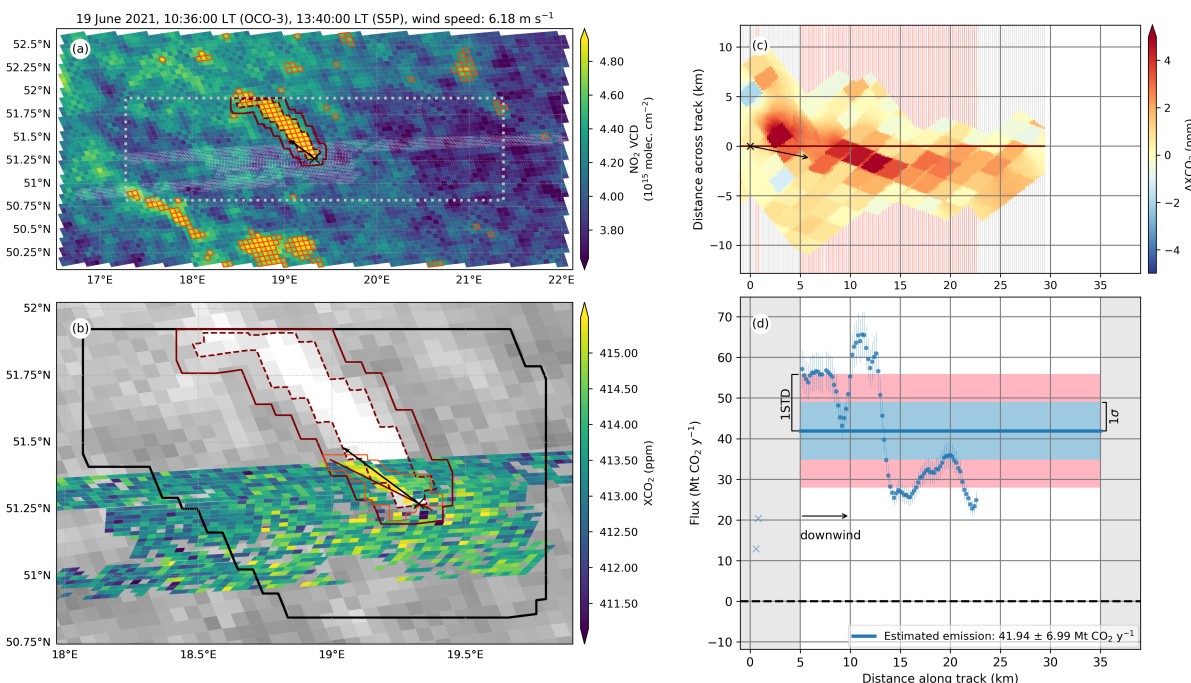

**Figure A1.** Overview of the top-down emission rate estimation steps for the scene on 19 June 2021. Analogous to Fig. 6 in the main text.



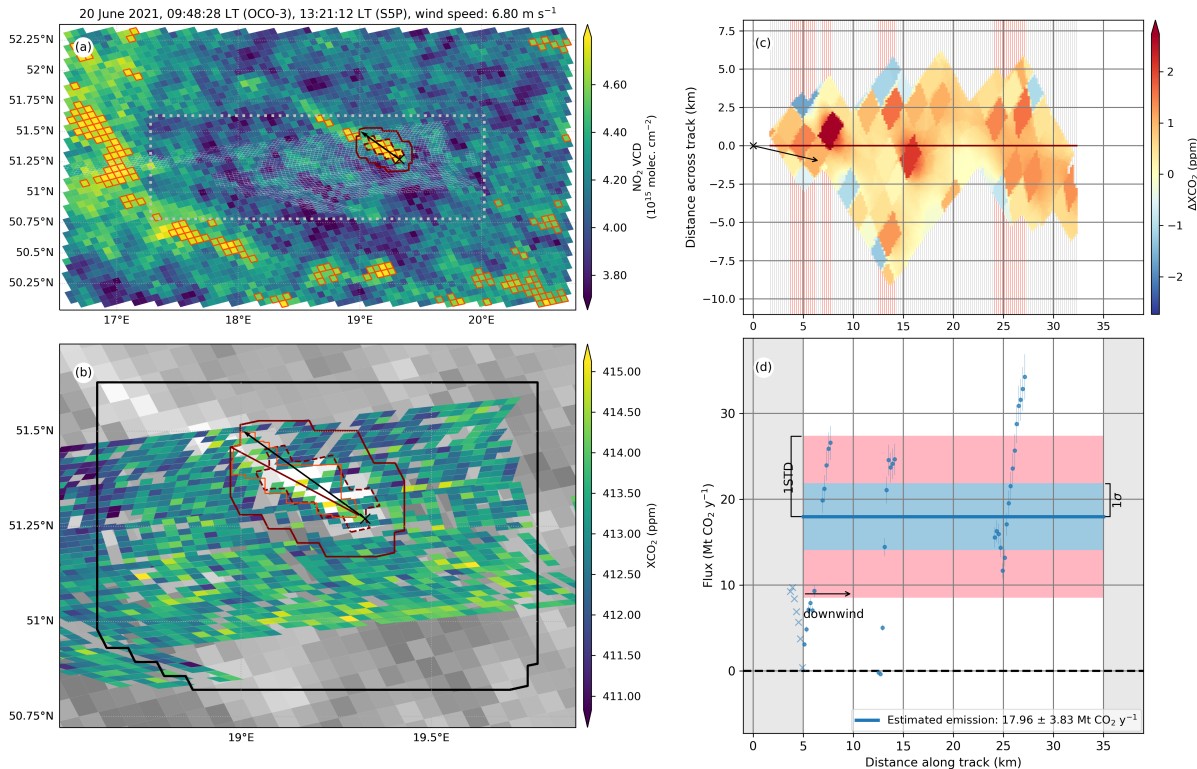

**Figure A2.** Overview of the top-down emission rate estimation steps for the scene on the 20 June 2021. Analogous to Fig. 6.

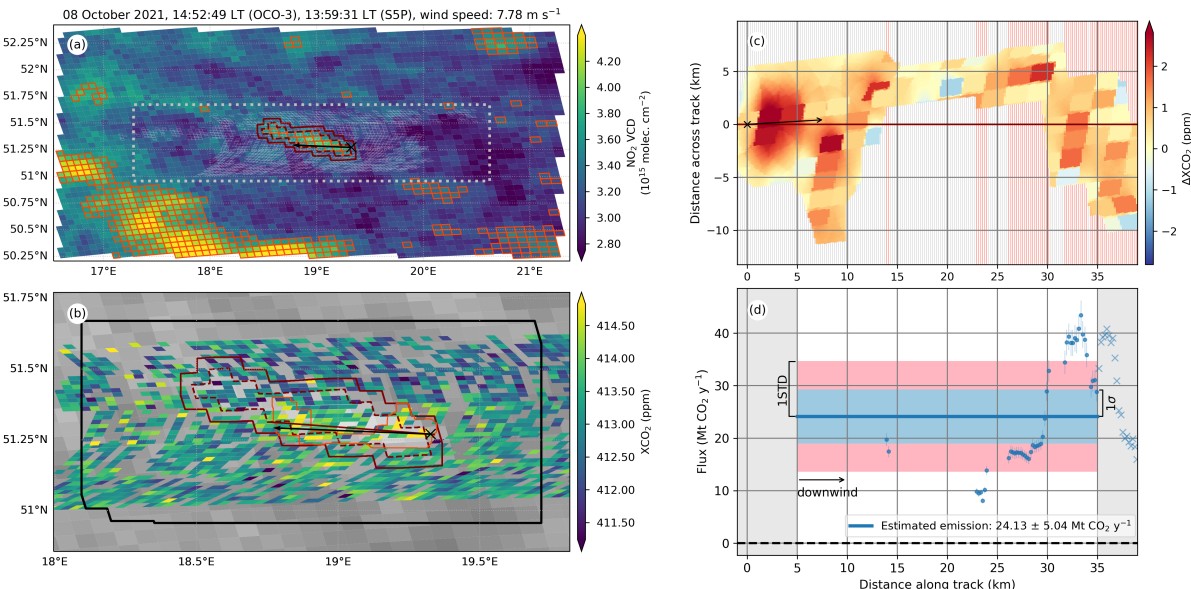

**Figure A3.** Overview of the top-down emission rate estimation steps for the scene on the 8 October 2021. Analogous to Fig. 6.





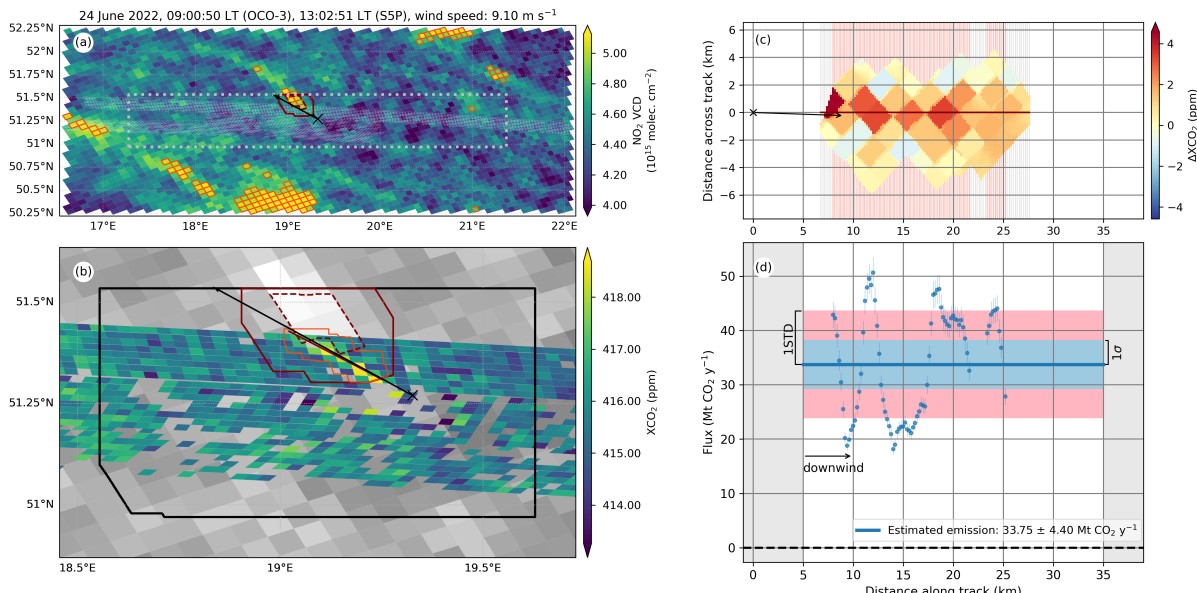

**Figure A4.** Overview of the top-down emission rate estimation steps for the scene on the 24 June 2022. Analogous to Fig. 6.



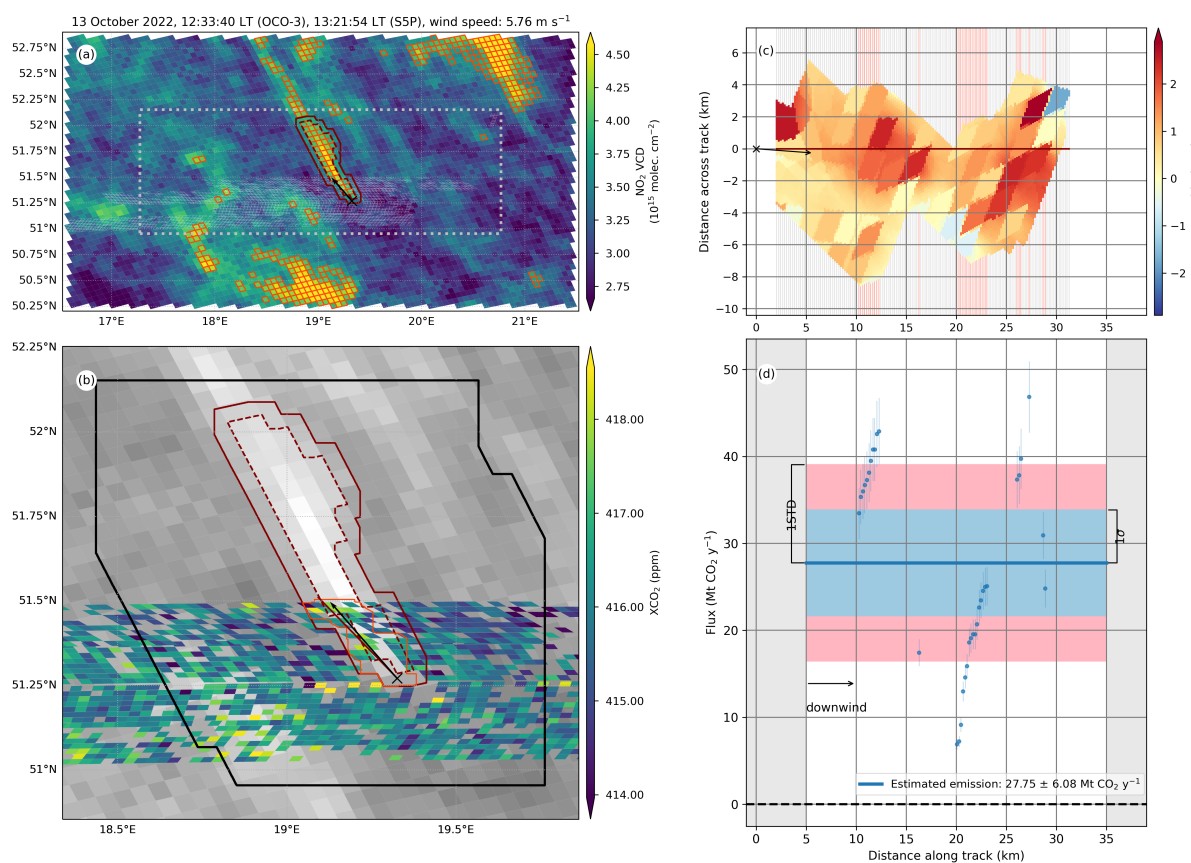

**Figure A5.** Overview of the top-down emission rate estimation steps for the scene on 13 October 2022. Analogous to Fig. 6.





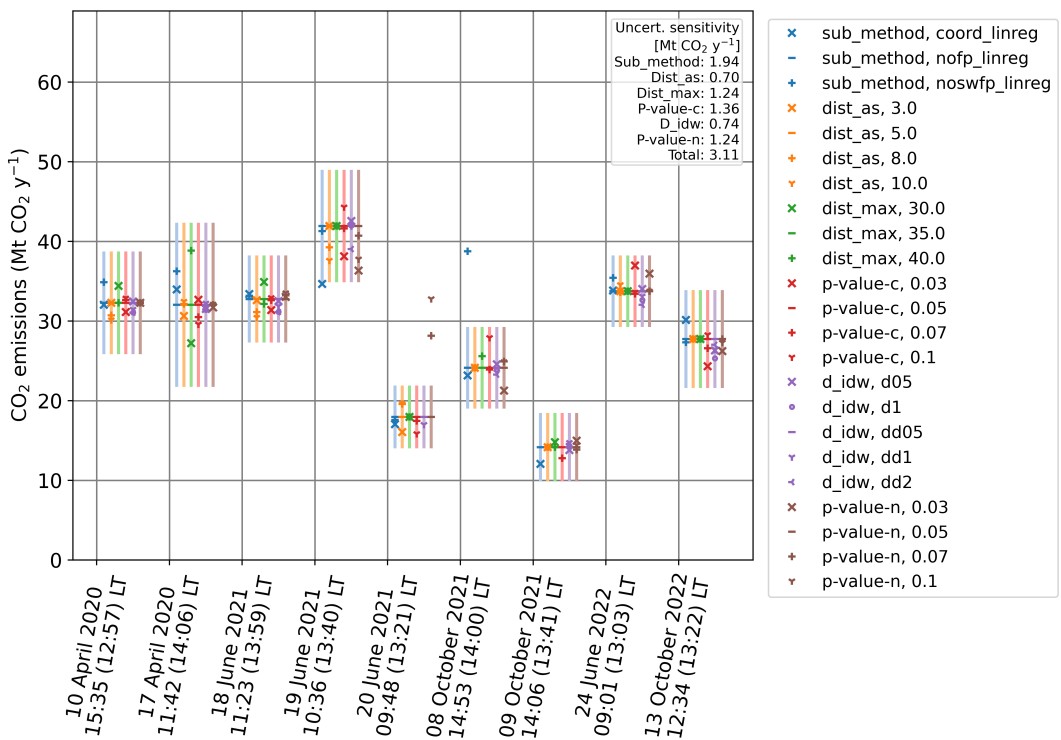

**Figure A6.** Sensitivity analysis performed to estimate a measure of the sensitivity uncertainty. The legend shows the names of the parameters taken into account and the values considered in each case: 1) p-values for the detection of the potential plume (`p-value-n`) and for plume refinement (`p-value-c`), taking values from 0.03 to 0.1 in both cases. 2) Area and weighting of the distance weighting interpolation (`d_idw`), where d indicates inverse distance weighting interpolation and dd squared inverse distance weighting interpolation, and the numbers refer to the radius of the used area, in tenths of degrees. 3) Limits of the plume range (`dist_as` and `dist_max` for the lower and upper thresholds, respectively). The values are in kilometres. 4) The function used to fit the $XCO_2$ background (`sub_method`): linear dependence on longitude and latitude with a possible swath bias (`nofp_linreg`), linear dependence on longitude and latitude with a possible swath and footprint biases (`coord_linreg`), only linear dependence on longitude and latitude (`noswfp_linreg`). Each data point stands for the result of the analysis using the value indicated by the marker for each considered parameter. The dashes stand for the values used for the main analysis. The bars show, for each scene, the total uncertainty obtained using the parameters selected for the main analysis.





**Figure A7.** Summary of results, analogous to Fig. 9, for the sensitivity tests: a) Rotation of the wind direction to match the plume track. b) No quality filtering of the $XCO_2$ data. c) Potential plume definition through a wedge downwind of the source instead of $NO_2$ VCD.





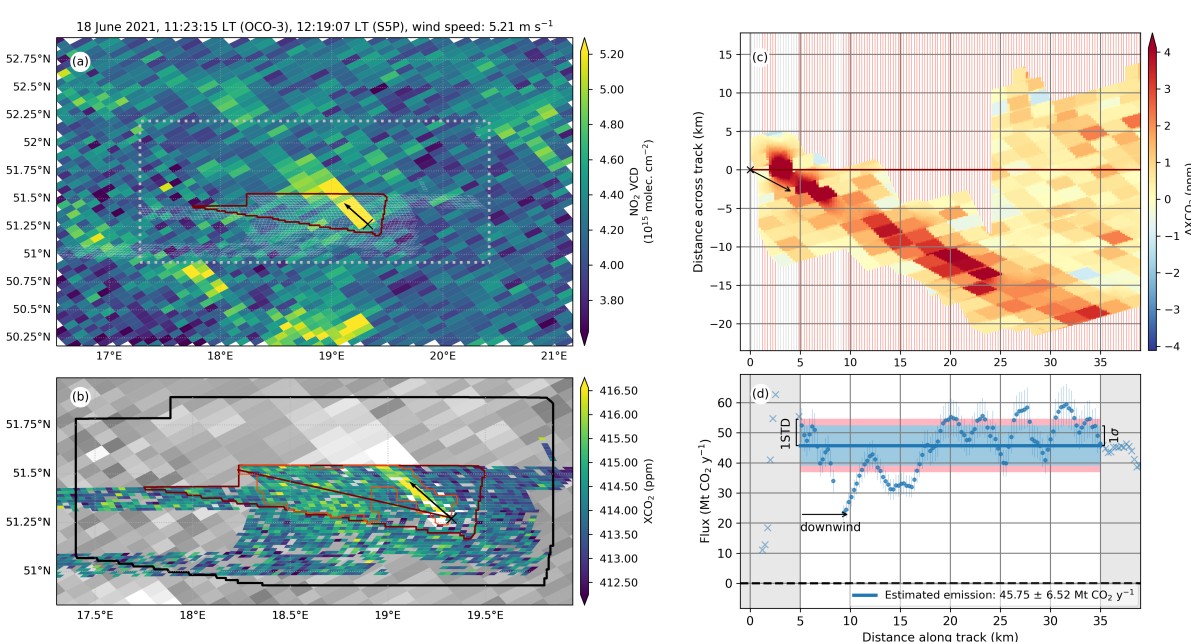

**Figure A8.** Overview of the top-down emission rate estimation steps for the scene on 18 June 2022. Analogous to Fig. 6. Instead on $NO_2$ data, a wedge downwind of the source was used to define the potential plume (as described in Sec. 2.4) $NO_2$.



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
