# Peer review of "A method for estimating localized $CO_2$ emissions from co-located satellite $XCO_2$ and $NO_2$ images"

_EGUsphere, 2023_

## Referee Comment (RC2)

Review of "A method for estimating localized CO2 emissions from co-located sdaellite XCO2 and NO2 images"
By Andrade et al, 2023, submitted to AMT

**Description and Reccomendation**
This paper describes a relatively automated plume-detection and plume flux inference using combined TROPOMI NO2 and OCO-3 XCO2 data. In this manuscript, they specifically apply their method to the Belchtow power plant in Poland, which is one of the largest coal-fired power plants in the world. They obtain generally good agreement between their inferred fluxes and those estimated by a standard bottom-up method (using data from the power plant itself to inform its instantaneous emissions).

They find generally good agreement with a completely different approach described recent in Nassar et al. (2022). Their approach uses a cross-sectional flux approach, wherein the emission plume is intersected with a number of cross-sections at varying distances downstream of the emitter; the Nassar et al. approach uses a Gaussian plume model fitting technique.

Overall, this manuscript is extremely well-written and gives an appropriately level of detail to a relatively complicated method with many steps and multiple data inputs. I recommend publication after my relatively minor comments have been addressed.

**General Comments**

As the authors point out, numerous papers have tried to use CO2 simultaneously with NO2 to quantify co2 emission rates from power plants. In principle, you can use CO2 alone (as Nasser at all, 2017, 2022 does) or NO2 alone (by assuming you know the emission rate). There as thus many ways to "mix and match" the information provided by the NO2 observations and the CO2 observations. A small discussion of the different assumptions one could make – and their associated "pros" and "cons" - would be most welcome somewhere in the paper, and what assumptions you personally chose. It appears that you (Page 6, line 148) only use NO2 to identify a region containing an emission plume – i.e. it is only used in a purely qualitative sense. Please clarify this in the paper. Do you use NO2 to localize the spot of the source? Or do you use the well-known coordinates of the Belchatow station? For instance, Hakkarainen et al (2023) in their "Building a Bridge" paper seem to simultaneously fit Gaussians to both the NO2 and CO2 CS's, and then use them to somehow construct an effective NOx-to-CO2 ratio, which is (somehow) used constrain the power plant CO2 flux. But even their paper is not very clear on this point.

It would be useful to expand your discussion of the pros and cons of your cross-sectional method (similar to that of Hakkarainen et al but more automated) and the more conventional Gaussian-plume model fit. Since the latter "fits all the data at once" it seems intuitively like it might avoid some of the errors your method is subject to, such as those caused by dispersion. But perhaps not – it seems very different to say. It implies an OSSE study with LES-modelgenerated plumes might be warranted, to investigate the pros and cons of these different techniques.

Comment on how much of this was automated vs. "Done by eye". It seems like the authors are trying to largely automate the technique but this is not entirely clear. A seemingly big advantage of this study is the amount of automation put into this work, such that it could potentially be applied to future sensors such as CO2M. I strongly recommend emphasizing the automation aspect of this work in the abstract.

**Specific Comments**

Sect 2.2: Your equations are all in terms of VCDs. You seem to be assuming locally flat ground. What is the topography in the region is significant? The VCD will generally scale proportionally to the surface pressure, assuming CO2 & NO2 are well mixed in the boundary layer. This will imprint topography on the VCD map. Can you please comment on how you account for this?

P8, L165: Will this NO2 smoothing potentially make the plume wider than it really is? Did you consider this effect, or can you comment on if it negligible?

P213, near L275: You should also mention that the wind speed & direction can easily vary over 1-3 hours. It seems like this is quite common and will certainly distort the plume and lead to flux inference errors. I would think this would lead to fairly coherent variability of the type that figure 5b exhibits. You take great pains to estimate the error due to variability in the source flux F; why not do the same for potential variations in the wind over the 1-3 hour period defining the plume length? At the very least, this should be commented on. Note that this is somewhat different than a mean bias error in the wind speed, such as you consider in section 2.3. Although perhaps this is inherently taken into account by your "dispersion uncertainty calculation", wherein you empirically estimate the covariance function from the semi-variogram of the data itself?

Section 2.3: I think you also should include a measurement uncertainty term. Currently there is no term that represents potential biases in the measurements, or if there is, I can't see it. According to Nassar et al's work, this term is not usually large, but warrants at least commenting on it in the paper.

Section 2.3.1 & Figure 9: It seems like your dispersion uncertainty can often be extremely large and dominate the overall uncertainty. The discussion in this section, however, is fairly technical and contains no plots that give the reader evidence that you've done this correctly. I think plotting semi-variograms and the overlaid fits would help, especially if you compared two very different cases, such as 17-April-2020 (very large dispersion uncertainty) and 20-June-2021 (small dispersion uncertainty).

Section 2.3.2: I'm not crazy about 0.5 m/s as a 1-sigma mean wind uncertainty. I prefer the Nassar et al (2017,2022) approach using MERRA-2 – ERA5, though it would be better to include

a floor of 0.5 for the reasons you state.  It would be nice if you could comment on this methodological difference in this subsection.

P25, L517:  The location of the lignite pit is poorly represented by the v10.4 OCO-3 digital elevation map, and leads to time-independent biases over those locations.  This general issue is discussed in Jacobs et al., 2023 (https://amt.copernicus.org/preprints/amt-2023-151/).  At the end of these comments, I've attached a figure made by the OCO team showing the effects of a change in DEM on this feature.  Therefore, I think it is fully warranted to filter out these spatial samples around the pit by hand, or the full case-filtering approach you describe.

**Technical Comments**

P2, L38: need "e.g." before Reuter et al, Nassar et al citations.  There have been many such publications and these are simply examples.  This is the case in many places throughout the paper.  Such as P2, L43; I doubt Ciais was the first author to note the long lifetime of CO2 in the atmosphere!  The rule I use is this.  If a paper was the first to say or show something you cite it directly.  If the paper is merely an example of many such papers saying the same thing, and was not the first, you need "e.g.".

P7, Fig3: It is VERY hard to see the "grey lines" of the SAM scan.  Please modify to make a little more visible.

P25, L515:  "in 0.3" → "by 0.3"

[Figure]

**Figure 1: Two sets of OCO-2 version 11 and version 11.1 retrievals of Target-mode observations near the Belchatow power plant.  The lignite pit to the southwest of the power plant (black circle) exhibits biases in both retrieval versions, though the use of the Copernicus DEM greatly reduces the size of the bias.  It is likely the Copernicus DEM is still not accurate enough for this area, which may have undergone anthropogenic changes in the surface since the Copernicus DEM data were acquired (roughly 10 years ago).  The prevailing wind direction of the plume is shown as a black arrow.   Cases identified by Ray Nassar at ECCC.**

---

## Author Comment (AC1)

**Point-by-point response to the reviewer's comments**

We thank the reviewer Ray Nassar for the comments and suggestions provided, which have helped to improve the quality of the manuscript. We have taken into account all the suggested improvements:

**Specific Points**

**Comment 1**:  Line 14: "possible thanks to" would better be rephrased as "made possible by"

**Reply**:  We rephrased "possible thanks to" as suggested.

**Comment 2**:  Line 60 and 61: capitalization of ENVISAT and TANSO is the advised, although TANSO-FTS is the complete name of the instrument.'

**Reply**:  We changed "Tanso" to "TANSO-FTS" and capitalized ENVISAT.

**Comment 3**:  65: A Gaussian plume model does not account for eddies, however, it relies on the reasonable assumption that their effects are negligible for multi-kilometer spatial scales. It is recommended that the sentence is expanded to clarify this fact.

**Reply**:  We expanded the sentence as suggested in L74 (of the revised version).

**Comment 4**:  Line 116: "instantaneous hourly" would be more informative than just "hourly" to distinguish from an hourly average value.

**Reply**:  We modified it accordingly.

**Comment 5**:  Figure 1 caption "gross" should be "cross" or X.

**Reply**:  We corrected the typo.

**Comment 6**:  Line 156: Is there any justification of the requirement of less than 5 hours? Obviously a shorter offset in time is better, but are there any studies to quantify the effect that might justify this value? Both wind speed and direction could change significantly over a period of 5 hours, as discussed later around line 190.

**Reply**: The requirement of less than 5 hours is a generous *ad hoc* criterion. Despite the fact that both wind speed and direction could change over this time period, which would lead to different observed plume shapes for NO2 and CO2, small changes in the wind speed and direction in the time between overpasses do not play a significant role for the application of our method. This is because NO2 data is only used to define a potential plume that serves as a bounding box, i.e. it only constrains a spacial region to find the CO2 plume. In addition, this potential plume is the result of extending the detected NO2 plume, which increases the likelihood of the real CO2 plume being contained within the detected potential plume. Therefore, as long as there are no drastic changes in the wind speed and direction in the time between both overpasses, as is the case for the analysed scenes, the potential plume detected using NO2 will most likely contain the CO2 plume. The scene on 18 June 2021 (Fig. 7) is the only counter example. In this scene, the CO2 plume seems to not be fully contained within the potential plume (as mentioned in L636 of the revised version), which has however just a small effect for the emission estimation because the part of the CO2 that we miss is mostly farther than 35 km downwind of the source.

Thanks to this comment we have seen an inaccuracy in the manuscript. In L615 (of the initial version) we had mentioned, referring to the scene on 18 June 2021, that "the part of the CO2 plume that we miss is beyond the plume range, having no effect on the final result". However, in Fig. 7 we can appreciate that we miss part of the plume after about 31 km, leading to a small underestimation of the emissions in this case. We have corrected this in L637 of the revised version.

For a more systematic analysis of other scenes and targets, more filters need to be developed to automatically discard scenes where the detected plume only partially contains the CO2 emission plume due to changes in the wind speed and/or direction.

**Comment 7**: Line 208: This approach to account for swath bias is interesting and likely contributes to an improvement in emission estimates, however, should the swath numbering be "j = 1,2,... n", rather than only going up to n-1? Is it n-1 since the first swath has no offset, so j = 0,1,2 ... n-1, where s0 = 0?

**Reply**: The reason for the swath bias numbering being "j = 1,2,... n - 1" instead of "j = 1,2,... n", for a total of $n$ swaths is that one swath has no offset because it was set as the reference. If we defined a $n$-th swath bias, $s_n$, Eq. 3 would lead to an under-determined linear regression without an

unique solution, i.e., the parameter $a_0$ could have any value and the different $s_j$ would adjust accordingly. We can solve this indetermination by setting $s_n = 0$ as a reference, without loss of generality.

**Comment 8**: Line 277: 1-3 hours for the characteristic time used to determine the bottom-up value is consistent with the findings of Nassar et al. (2021, https://doi.org/10.1016/j.rse.2021.112579, e.g. Figure 1 and sec 2.5), which considered the plume extent, time since emissions to derive a time-weighted or 'dynamic' bottom-up value. This similar analysis is worth mentioning very briefly and citing.

**Reply**: We mentioned the similarity of both approaches and cited Nassar et al. (2021).

**Comment 9**: Section 2.3, uncertainty. Is there any uncertainty related to the observations? It was not entirely clear to me if this was indirectly included in the dispersion or sensitivity terms. The sensitivity term does account for uncertainty in the observations for background, but not necessarily the plume. Can the authors clarify?

**Reply**: We did not explicitly add an extra term related to the observations. We assumed that any biases in the plume observations are removed with the background subtraction. We clarified this in L314 (of the revised version) by adding the following comment: "We did not explicitly consider a XCO2 measurement error under the assumption that, at the relatively small spatial scales of the analysed scenes, any bias in the XCO2 data is corrected for when subtracting the background. Random errors in the XCO2 values are included in the dispersion uncertainty."

**Comment 10**: Line 559: "lead" should be "led"

**Reply**: Corrected.

**Comment 11**: Line 595: It is not surprising that the difference between applying quality filters and ignoring them reduced when observations near the Bełchatów lignite pit were excluded. The digital elevation model for OCO-3 v10 data does not account for recent anthropogenic effects on topography such as this, so biased XCO2 data will result through erroneous surface pressures. Although no DEM will be perfectly up to date with respect to anthropogenic effects on topography, the Copernicus DEM which

will be used in OCO-3 v11 data will reduce the problem and thus the difference between quality-filtering and not, will be reduced.

**Reply**: Indeed, it is not surprising that quality filtering reduces the observations near the lignite pit. As Christopher O'Dell suggested in their review to this manuscript, it might be warranted to filter out data points around the pit by hand because of the reason you state. Using a more accurate Digital Elevation Model (DEM) will presumably help to reduce the problem.

---

## Author Comment (AC2)

**Point-by-point response to the reviewer's comments**

We thank the reviewer Christopher O'Dell for the comments and suggestions provided, which have improved the quality of the manuscript. We have considered all the suggested improvements:

**General comments**

**Comment 1**: As the authors point out, numerous papers have tried to use CO2 simultaneously with NO2 to quantify CO2 emission rates from power plants. In principle, you can use CO2 alone (as Nasser at all, 2017, 2022 does) or NO2 alone (by assuming you know the emission rate). There as thus many ways to "mix and match" the information provided by the NO2 observations and the CO2 observations. A small discussion of the different assumptions one could make – and their associated "pros" and "cons" - would be most welcome somewhere in the paper, and what assumptions you personally chose. It appears that you (Page 6, line 148) only use NO2 to identify a region containing an emission plume – i.e. it is only used in a purely qualitative sense. Please clarify this in the paper. Do you use NO2 to localize the spot of the source? Or do you use the well-known coordinates of the Belchatow station? For instance, Hakkarainen et al (2023) in their "Building a Bridge" paper seem to simultaneously fit Gaussians to both the NO2 and CO2 CS's, and then use them to somehow construct an effective NOx-to-CO2 ratio, which is (somehow) used constrain the power plant CO2 flux. But even their paper is not very clear on this point.

**Reply**: A small discussion of the different approaches to quantify CO2 emissions from observations of CO2 alone, NO2 alone and a combination of both was included in the introduction (L44-63 of the revised version).

A clarification of our use of NO2 to only identify a region containing an emission plume was added in L61.

The coordinates of the Belchatow power station are considered well-known and used as an input. This was clarified in L160 (of the revised version).

**Comment 2**: It would be useful to expand your discussion of the pros and cons of your cross-sectional method (similar to that of Hakkarainen et al but more automated) and the more conventional Gaussian-plume model fit. Since the latter "fits all the data at once" it seems intuitively like it might avoid some of the errors your method is subject to, such as those caused

by dispersion. But perhaps not – it seems very different to say. It implies an OSSE study with LES-model-generated plumes might be warranted, to investigate the pros and cons of these different techniques.

**Reply**: A slightly expanded discussion of pros and cons of a cross-sectional flux method vs a Gaussian-plume model fit was included from L74 (of the revised version) onwards.

In a cross-sectional flux method, if the emission rate is computed from the mean of a set on independent cross-sectional fluxes, any random errors, in theory, average out. On the other hand, a Gaussian-plume model fit is based on the assumption of a steady state and therefore neglects turbulent plume structures. However, if the plume deviates from this steady-state behaviour, as is often the case at daytime (as seen e.g. in simulated plumes, Brunner et al, 2023, Figs. 2, 3) a Gaussian model might fail to describe the real plume structure. Therefore, in such cases, fitting "all the data at once" might not have any advantages over studying a set of cross-sectional fluxes, because there is no apparent reason why emission estimates obtained though the mean of independent cross-sectional fluxes would contain more errors than those obtained by means of a gaussian fit. Our cross-sectional flux method has the additional advantage of not assuming a plume shape or behaviour other than weak stationarity, allowing us to estimate an uncertainty associated to intra-plume structures from the analysis of the cross-sectional fluxes dispersion. On the other hand, the emission estimate uncertainty obtained from a Gaussian fit is likely underestimated if the deviation from the steady-state assumption is not included in it.

Therefore, not considering an overall Gaussian fit does not in theory imply more errors in the results, but the possibility to quantify them. However, as suggested, an OSSE study with LES-model-generated plumes would probably help to clarify this point.

**Comment 3**: Comment on how much of this was automated vs. "Done by eye". It seems like the authors are trying to largely automate the technique but this is not entirely clear. A seemingly big advantage of this study is the amount of automation put into this work, such that it could potentially be applied to future sensors such as CO2M. I strongly recommend emphasizing the automation aspect of this work in the abstract.

**Reply**: Thank you for the suggestion. We explicitly mentioned the automation of the method in the abstract and in the method description.

In principle, besides the input datasets illustrated in Fig. 2, the coordinates of the source and the parameters mentioned in the description of

the method are used as an input. Additionally, the bottom-up CO2 intensity was manually computed and given as an input. That all set up, the described analysis runs automatically for all the available scenes within the selected time period.

**Specific Comments**

**Comment 4**: Sect 2.2: Your equations are all in terms of VCDs. You seem to be assuming locally flat ground. What is the topography in the region is significant? The VCD will generally scale proportionally to the surface pressure, assuming CO2 and NO2 are well mixed in the boundary layer. This will imprint topography on the VCD map. Can you please comment on how you account for this?

**Reply**: The elevation in analysed region varies typically less than 75 m, having maximum elevation differences in the order of 200 m. VCD maps do contain this imprinted topography. For the potential plume detection (using NO2 data), we can neglect this effect for the small height difference about the Belchatow power plant. We can illustrate that as follows. Let us consider the VCD with respect to altitudes $z$ and $z + \Delta z$. Assuming that NO2 is well mixed in the boundary layer and hydrostatic balance, the number of air molecules exponentially decreases with altitude, $\text{VCD}(z+\Delta z) = \text{VCD}(z)\, e^{-\Delta z/H}$, where $H \approx 8.5$ km is the atmospheric scale height. The effect of topography for individual VCD considering $\Delta z = 0.2$ km would lead to $\text{VCD}(z)$- $\text{VCD}(z + \Delta z) \approx 0.02\, \text{VCD}(z)$. Therefore, the effect of topography for this scene is about 2 orders of magnitude smaller than VCD, while typical VCD errors and background standard deviation are about 0.2-0.3 $\text{VCD}(z)$. Since we use NO2 VCD only for the detection of a potential plume though a statistical test, and the NO2 VCD is much larger than the topography effect, the NO2 VCD error is the deciding factor when selecting the enhanced NO2 VCD.

For the computation of the cross-sectional fluxes, we need the VCD containing the imprinted topography. We converted the XCO2 anomaly to VCD through $n_d$ (the number of dry air molecules per unit area). This $n_d$ was obtained from ERA5 at the centre coordinates of each OCO-3 pixel taking the Earth's surface altitude from the L2 Lite OCO-3 product. Therefore, the topography of the region was considered to compute the VCD from the XCO2 anomaly.

**Comment 5**: P8, L165: Will this NO2 smoothing potentially make the plume wider than it really is? Did you consider this effect, or can you comment on if it negligible?

**Reply**: NO2 smoothing can make the plume wider. A widening of the NO2 detected plume would translate into less constrainment of the region to detect the CO2 plume.

This widening is negligible if the widened plume does not include emission plumes arising from other sources or other enhanced XCO2 background structures of about the same spatial extent as the CO2 plume. The widening becomes relevant if there are other emission sources or background XCO2 structures close to the emission plume, since they are more likely to be included in the potential plume, leading to a potential inclusion of these XCO2 enhancements in the CO2 detected plume and consequently to an overestimation of the emission rate. The size of the widening is dependent on the TROPOMI pixel size and has at most the diametral size of two TROPOMI pixels. Because TROPOMI pixels become larger in the across-flight direction, this widening is more pronounced towards the edges of the TROPOMI swath. That is why, in the scene selection procedure, in order to minimize this effect, we discarded scenes for which a large fraction of the TROPOMI pixels belonged to the most off-nadir pixels of the swath.

**Comment 6**: P213, near L275: You should also mention that the wind speed and direction can easily vary over 1-3 hours. It seems like this is quite common and will certainly distort the plume and lead to flux inference errors. I would think this would lead to fairly coherent variability of the type that figure 5b exhibits. You take great pains to estimate the error due to variability in the source flux F; why not do the same for potential variations in the wind over the 1-3 hour period defining the plume length? At the very least, this should be commented on. Note that this is somewhat different than a mean bias error in the wind speed, such as you consider in section 2.3. Although perhaps this is inherently taken into account by your "dispersion uncertainty calculation", wherein you empirically estimate the covariance function from the semi-variogram of the data itself?

**Reply**: Any oscillations in the wind speed and direction about their value at overpass time are considered in the dispersion uncertainty. So is, to a certain extent, a drift in the wind speed. A drift in the wind speed would translate into a drift in the obtained cross-sectional fluxes, which would lead to a larger empirical semivariances (whose increase is larger for larger lags) and therefore a larger dispersion uncertainty. Therefore, we can assume

that small drifts in the wind speed (i.e. those which would lead to drifts in the cross-sectional fluxes that are much smaller than their oscillations along the plume track) are taken into account in the dispersion uncertainty. For example, as mentioned in L576-584 (of the revised version), for a relatively large decrease by 1 m s$^{-1}$ in the wind speed along the plume track, from a typical wind speed of about 6 m s$^{-1}$ and emissions of about 30 MtCO$_2$ y$^{-1}$, a drift by about 5 MtCO$_2$ y$^{-1}$ would be expected for individual cross-sectional fluxes, which is much smaller than the oscillations observed. However, a significant drift in the wind vector would mean a departure from the weak-stationarity assumption, under which we have computed the mean emission estimate and its uncertainty from the semivariogram.

**Comment 7**: Section 2.3: I think you also should include a measurement uncertainty term. Currently there is no term that represents potential biases in the measurements, or if there is, I can't see it. According to Nassar et al's work, this term is not usually large, but warrants at least commenting on it in the paper.

**Reply**: We added the following comment on this in L314 (of the revised version): "We did not explicitly consider a XCO2 measurement error under the assumption that, at the relatively small scales of the analysed scenes, any bias in the XCO2 data is corrected for when subtracting the background. Random errors in the XCO2 values are included in the dispersion uncertainty."

**Comment 8**: Section 2.3.1 and Figure 9: It seems like your dispersion uncertainty can often be extremely large and dominate the overall uncertainty. The discussion in this section, however, is fairly technical and contains no plots that give the reader evidence that you've done this correctly. I think plotting semi-variograms and the overlaid fits would help, especially if you compared two very different cases, such as 17-April-2020 (very large dispersion uncertainty) and 20-June-2021 (small dispersion uncertainty).

**Reply**: A new figure, Fig. A9, showing the semi-variograms for all analysed scenes was added to Appendix A.

The low uncertainty on 20 June 2021 is, as discussed in L583 (of the revised version), probably an underestimation due to low number of valid CSs with large gaps without information.

**Comment 9**: Section 2.3.2: I'm not crazy about 0.5 m/s as a 1-sigma mean wind uncertainty. I prefer the Nassar et al (2017,2022) approach

using MERRA-2 – ERA5, though it would be better to include a floor of 0.5 for the reasons you state. It would be nice if you could comment on this methodological difference in this subsection.

**Reply**:

Taking an ensemble approach of only two products (MERRA-2 and ERA5) might be insufficient to characterize the true uncertainty or the wind speed, since it will be affected by random errors from both products and thus also oscillate randomly, not necessarily representing a true uncertainty.

Additionally, MERRA-2 provides 3h averages of the wind vector while ERA 5 winds are hourly instantaneous values. Because of this, MERRA-2 winds are likely to have less variability that ERA5 winds, so the difference MERRA-2 – ERA5 might overestimate the uncertainty.

We included a mention of the methodological difference in Sec. 2.3.2.

**Comment 10**:   P25, L517: The location of the lignite pit is poorly represented by the v10.4 OCO-3 digital elevation map, and leads to time-independent biases over those locations. This general issue is discussed in Jacobs et al., 2023 (https://amt.copernicus.org/preprints/amt-2023-151/). At the end of these comments, I've attached a figure made by the OCO team showing the effects of a change in DEM on this feature. Therefore, I think it is fully warranted to filter out these spatial samples around the pit by hand, or the full case-filtering approach you describe.

**Reply**:  Thank you for this information. We have not yet performed any detailed study on DEM effects, but using an accurate DEM will reportedly help to reduce biases in the XCO2 product and in the emission estimate when using this product.

**Technical Comments**

**Comment 11**:   P2, L38: need "e.g." before Reuter et al, Nassar et al citations. There have been many such publications and these are simply examples. This is the case in many places throughout the paper. Such as P2, L43; I doubt Ciais was the first author to note the long lifetime of CO2 in the atmosphere! The rule I use is this. If a paper was the first to say or show something you cite it directly. If the paper is merely an example of many such papers saying the same thing, and was not the first, you need "e.g.".

**Reply**: That is a reasonable rule. We changed that accordingly at several places throughout the paper.

**Comment 12**: P7, Fig3: It is VERY hard to see the "grey lines" of the SAM scan. Please modify to make a little more visible.

**Reply**: We made the grey lines more visible in Figs.3-8 and Figs. A1-A5.

**Comment 13**: P25, L515: "in 0.3" → "by 0.3"

**Reply**: We corrected "in" to "by".